# A *Drosophila* model of neuronal ceroid lipofuscinosis *CLN4* reveals a hypermorphic gain of function mechanism

Elliot Imler[1,2], Jin Sang Pyon[2,3], Selina Kindelay[2,3], Meaghan Torvund[1,2], Yong-quan Zhang[4,5], Sreeganga S Chandra[4,5], Konrad E Zinsmaier[2,6]*

[1]Graduate Interdisciplinary Program in Neuroscience, University of Arizona, Tucson, United States; [2]Department of Neuroscience, University of Arizona, Tucson, United States; [3]Undergraduate Program in Neuroscience and Cognitive Science, Department of Molecular and Cellular Biology, University of Arizona, Tucson, United States; [4]Department of Neuroscience, Yale University, New Haven, United States; [5]Department of Neurology, Yale University, New Haven, United States; [6]Department of Molecular and Cellular Biology, University of Arizona, Tucson, United States

**Abstract** The autosomal dominant neuronal ceroid lipofuscinoses (NCL) *CLN4* is caused by mutations in the synaptic vesicle (SV) protein CSPα. We developed animal models of *CLN4* by expressing *CLN4* mutant human CSPα (hCSPα) in *Drosophila* neurons. Similar to patients, *CLN4* mutations induced excessive oligomerization of hCSPα and premature lethality in a dose-dependent manner. Instead of being localized to SVs, most *CLN4* mutant hCSPα accumulated abnormally, and co-localized with ubiquitinated proteins and the prelysosomal markers HRS and LAMP1. Ultrastructural examination revealed frequent abnormal membrane structures in axons and neuronal somata. The lethality, oligomerization and prelysosomal accumulation induced by *CLN4* mutations was attenuated by reducing endogenous wild type (WT) dCSP levels and enhanced by increasing WT levels. Furthermore, reducing the gene dosage of Hsc70 also attenuated *CLN4* phenotypes. Taken together, we suggest that *CLN4* alleles resemble dominant hypermorphic gain of function mutations that drive excessive oligomerization and impair membrane trafficking.
DOI: https://doi.org/10.7554/eLife.46607.001

*For correspondence:
kez4@email.arizona.edu

**Competing interests:** The authors declare that no competing interests exist.

## Introduction

NCLs comprise a group of progressive neurodegenerative diseases with 14 known disease-associated genes, termed *CLN1-14* (*Haltia, 2003*; *Haltia and Goebel, 2013*; *Jalanko and Braulke, 2009*; *Mole and Cotman, 2015*). NCLs have mostly an infantile or juvenile symptomatic onset and are characterized by loss of vision, gait abnormalities, seizures, dementia, and premature death. In general, NCLs are considered lysosomal storage diseases due the accumulation of lipofuscin and are typically caused by recessive loss of function mutations with one exception, *CLN4*. Accordingly, lysosomal dysfunction, dysregulated ER-lysosomal trafficking, or aberrant lipid modifications are thought to be the basis for most NCLs, consistent with the known function of the mutated genes (*Bennett and Rakheja, 2013*; *Cárcel-Trullols et al., 2015*; *Mole and Cotman, 2015*; *Warrier et al., 2013*). Understanding disease mechanisms of NCLs has implications as well for more prevalent diseases since mutations in a growing number of *CLN* genes also cause other diseases like frontotemporal lobar degeneration, progressive epilepsy with mental retardation, spinocerebellar ataxia, retinitis pigmentosa, juvenile cerebellar ataxia, or Parkinson disease 9 (*Bras et al., 2012*; *Mole and Cotman, 2015*; *Yu et al., 2010*).

The autosomal dominantly inherited NCL *CLN4* has an adult onset between 25 and 46 years. *CLN4* is caused by either the amino acid (aa) substitution L115R or the single amino acid deletion L116Δ in the SV protein CSPα, which is encoded by the human *DNAJC5* gene (*Benitez et al., 2011*; *Cadieux-Dion et al., 2013*; *Nosková et al., 2011*; *Velinov et al., 2012*). CSPα is unique among NCL-associated genes since it encodes a SV protein with no known lysosome-associated functions. Accordingly, there is no *CLN4* model explaining lysosomal failure.

CSPα is an evolutionary conserved neuroprotective co-chaperone of Hsc70 and required to maintain synaptic function and prevent neurodegeneration (*Burgoyne and Morgan, 2011*; *Burgoyne and Morgan, 2015*; *Zinsmaier, 2010*). Gene deletions in flies and mice cause progressive locomotor defects, paralysis and premature death due to neurodegeneration (*Chandra et al., 2005*; *Fernández-Chacón et al., 2004*; *Umbach et al., 1994*; *Zinsmaier, 2010*; *Zinsmaier et al., 1994*). On SVs, CSPα forms a molecular chaperone complex with Hsc70 for a selected set of clients, which include SNARE proteins and dynamin (*Chandra et al., 2005*; *Nie et al., 1999*; *Sharma et al., 2012*; *Sharma et al., 2011*; *Zhang et al., 2012*). Maintaining SNARE and dynamin function is likely key to CSP's neuroprotective role (*Burgoyne and Morgan, 2011*; *Rozas et al., 2012*; *Sharma et al., 2012*; *Sharma et al., 2011*).

The *CLN4* causing dominant mutations L115R and L116Δ are clustered in the palmitoylated cysteine-string (CS) domain of CSPα, which mediates CSPα's secretory trafficking to axon terminals, its SV association, and its dimerization (*Arnold et al., 2004*; *Chamberlain and Burgoyne, 1998*; *Greaves and Chamberlain, 2006*; *Greaves et al., 2008*; *Ohyama et al., 2007*; *Stowers and Isacoff, 2007*; *Swayne et al., 2003*). Palmitoylation of the CS domain enables CSPα's export from the ER and Golgi (*Chamberlain and Burgoyne, 1998*; *Greaves and Chamberlain, 2006*; *Greaves et al., 2008*; *Ohyama et al., 2007*; *Stowers and Isacoff, 2007*). Palmitoylation must then be maintained for CSPα's association with synaptic vesicle precursors (SVPs) and/or SVs, presumably to due to the short lifetime of palmitoylation (*Fukata and Fukata, 2010*). The latter has been indicated by much reduced synaptic levels of CSP in loss of function mutants of the synaptic palmitoyl-transferase HIP14/DHHC17 (*Ohyama et al., 2007*; *Stowers and Isacoff, 2007*). Notably, there is a link between CSPα's degree of lipidation and lysosomal dysfunction. In a lysosomal disease mouse model of Mucopolysaccharidosis type IIIA (MPS-IIIA), palmitoylation of CSPα was decreased and its proteasomal degradation was increased (*Sambri et al., 2017*). Since overexpression (OE) of CSPα in MPS-IIIA mice ameliorated their presynaptic defects, neurodegeneration, and prolonged survival, CSPα could be a critical factor for the progression of many lysosomal diseases (*Sambri et al., 2017*).

Post-mortem analysis of *CLN4* patient brains suggests that dominant *CLN4* mutations have two key pathological effects: to reduce monomeric levels of lipidated CSPα and promote the formation of high-molecular weight CSPα protein aggregates/oligomers that are ubiquitinated (*Greaves et al., 2012*; *Henderson et al., 2016*; *Nosková et al., 2011*). Similar effects of the mutations were seen in HEK293T, PC12 cells, and fibroblasts from *CLN4* carriers (*Benitez and Sands, 2017*; *Greaves et al., 2012*; *Zhang and Chandra, 2014*). In vitro, *CLN4* mutant CSPα aggregates form in a time-, concentration- and temperature-dependent manner (*Zhang and Chandra, 2014*). Palmitoylation of CSPα promotes aggregation (*Greaves et al., 2012*), although it is not essential (*Zhang and Chandra, 2014*). In addition, post-mortem brains of *CLN4* patients exhibit large scale changes in protein palmitoylation (*Henderson et al., 2016*). *CLN4* mutations have no adverse short-term effects on CSPα's co-chaperone functions in vitro, including activation of Hsc70's ATPase activity, or the binding to chaperone clients like SNAP25 and dynamin (*Zhang and Chandra, 2014*). However, during prolonged incubation, the ability of mutant CSPα to stimulate Hsc70's ATPase activity declines considerably (*Zhang and Chandra, 2014*).

Several mechanisms of *CLN4*-induced neurodegeneration have been suggested. As with other neurodegenerative diseases, progressive aggregation of mutant CSPα alone may account for the dominant nature of *CLN4* mutations (*Greaves et al., 2012*; *Henderson et al., 2016*; *Nosková et al., 2011*; *Zhang et al., 2012*). In turn, this may progressively deplete total CSPα levels such that it causes a haplo-insufficiency that triggers neurodegeneration (*Greaves et al., 2012*; *Nosková et al., 2011*; *Zhang and Chandra, 2014*). Consistent with this possibility, one-year old heterozygous CSPα KO mice develop a moderate motor defect that is characterized by a reduced ability to sustain motor unit recruitment during repetitive stimulation (*Lopez-Ortega et al., 2017*). Alternatively, aggregating CSPα may have a dominant-negative effect and sequester WT CSPα (*Greaves et al., 2012*; *Henderson et al., 2016*; *Zhang et al., 2012*). However, the link between *CLN4* mutations,

lysosomal dysfunction and neurotoxicity remains unclear. Therefore, an animal model may provide critical insight into molecular and cellular mechanisms underlying *CLN4* disease.

Here, we generated two *Drosophila* models for *CLN4* by expressing either *CLN4* mutant hCSPα or dCSP in fly neurons. The humanized fly model has the unique advantage of being able to differentially visualize *CLN4* mutant hCSPα and WT dCSP. We show that both fly models replicate all key pathogenic biochemical properties of *CLN4,* including decreased monomeric CSP levels, increased levels of high-molecular weight, and ubiquitinated CSP oligomers. Further analysis revealed novel insights into mechanisms underlying *CLN4* pathology.

## Results

### Generation of a *Drosophila CLN4* model

We generated a *Drosophila* model of *CLN4* by expressing the disease-causing human proteins hCSPα-L115R, hCSPα-L116Δ (*Figure 1A*; denoted as L115 and L116 from now on) and the corresponding WT hCSPα control from a common genomic phi31-attP insertion site under the transcriptional control of the yeast Gal4-UAS expression system (*Brand and Perrimon, 1993*). Unless

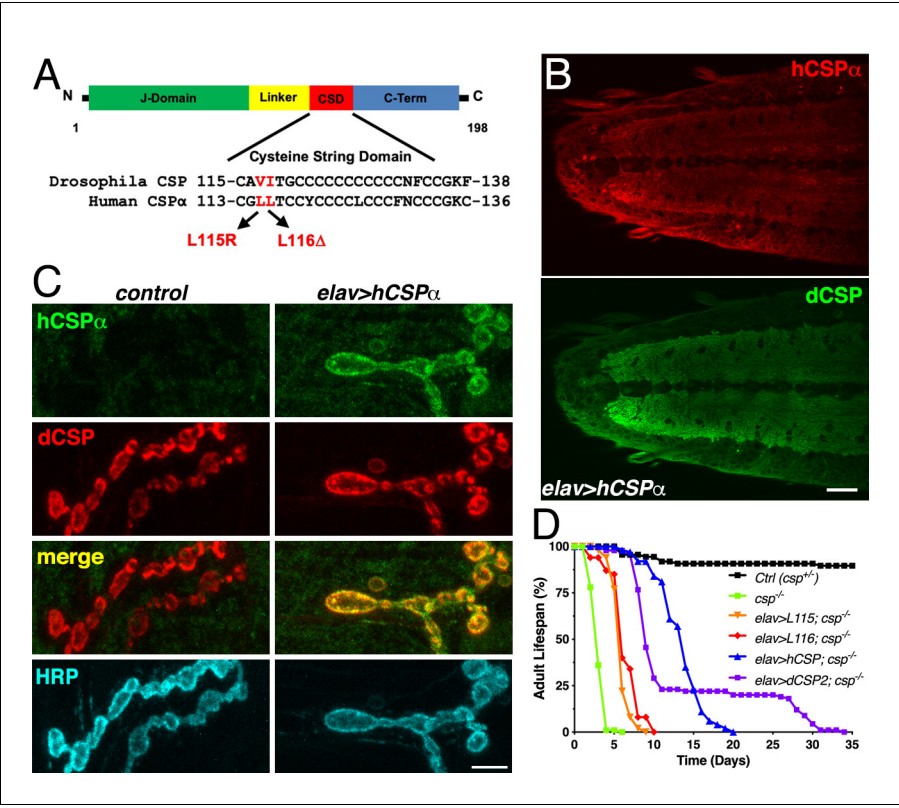

**Figure 1.** Generation of a *Drosophila CLN4* model. (**A**) Structure of CSP and position of *CLN4* mutations in the cysteine-string (CS) domain of hCSPα and dCSP. CSP's N-terminal J domain, linker domain and C-terminus are indicated. (**B**) Larval VNC of animals expressing WT hCSPα in neurons from a single transgene with an elav driver immunostained for hCSPα and endogenous dCSP. Scale bar, 20 μm. (**C**) Larval NMJs of control and animals expressing WT hCSPα immunostained for hCSPα, dCSP, and HRP marking the presynaptic plasma membrane. Scale bar, 5 μm. (**D**) Adult lifespan of control (*dcsp*$^{X1/+}$, black), *csp*$^{X1/R1}$ deletion mutants (green) and *csp*$^{X1/R1}$ mutants expressing WT hCSPα (blue), hCSP-L115, (orange), hCSPα-L116 (red), or dCSP2 (purple) with an elav driver.

DOI: https://doi.org/10.7554/eLife.46607.002

The following figure supplement is available for figure 1:

**Figure supplement 1.** Phenotypic effects of *CLN4* mutations.
DOI: https://doi.org/10.7554/eLife.46607.003

otherwise indicated, we used the pan-neuronal elav-Gal4 driver C155 (*Lin and Goodman, 1994*) to express these proteins exclusively in otherwise WT neurons ($w^{1118}$).

To evaluate whether WT hCSPα is at least partially functional in fly neurons, we first determined whether neuronally expressed hCSPα is properly palmitoylated and targeted to SVs. This analysis was aided by the availability of species-specific antibodies that discriminate between hCSPα and dCSP (*Figures 1B–C* and *2A*; *Figure 1—figure supplement 1A*). Neuronally expressed hCSPα was efficiently trafficked to axon terminals and co-localized with endogenous dCSP in the neuropil of the larval ventral nerve cord (VNC; *Figure 1B*) and larval neuromuscular junctions (NMJs; *Figure 1C*). Consistent with the normal trafficking of hCSPα, the majority of neuronally expressed hCSPα was palmitoylated, which was confirmed by treating larval protein extracts with 0.5 M hydroxylamine (*Figure 1—figure supplement 1A*).

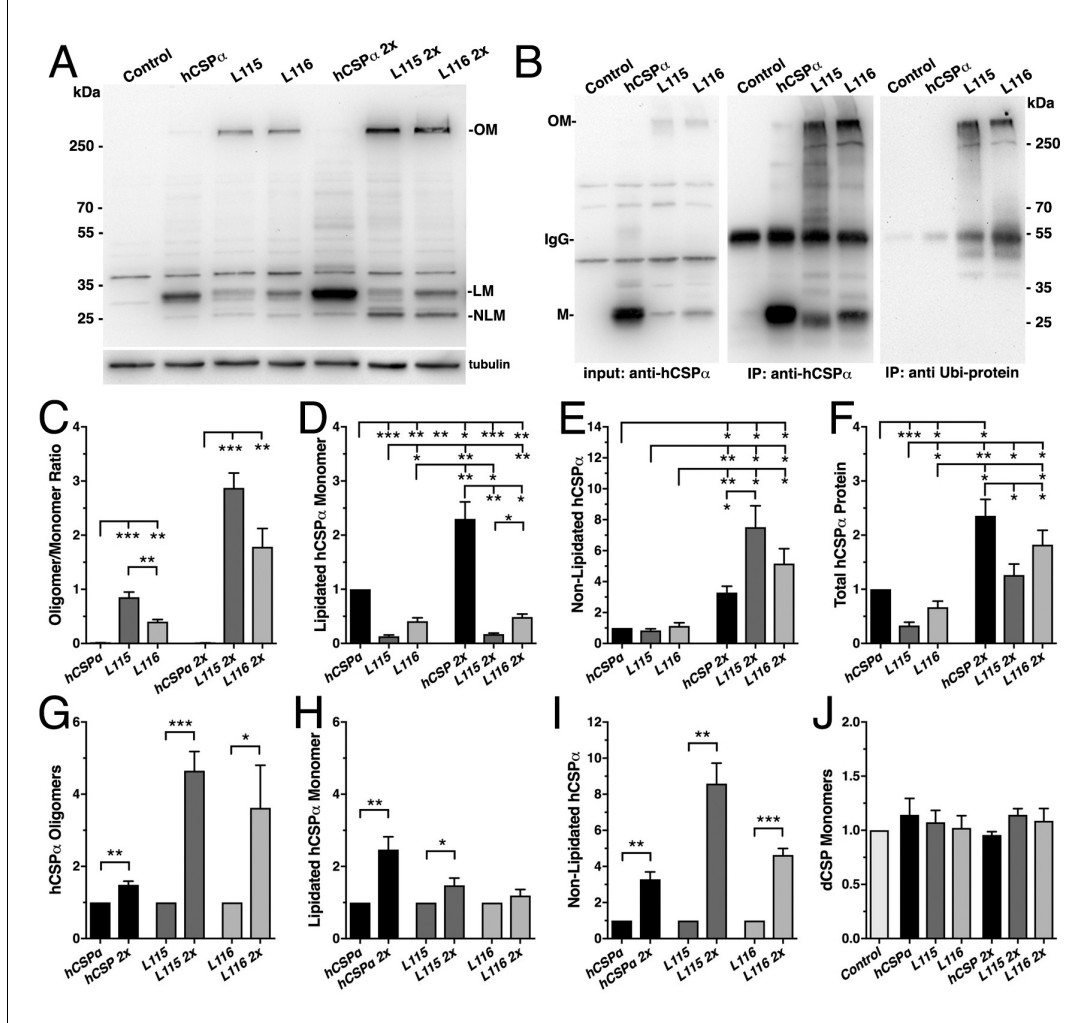

**Figure 2.** *CLN4* mutations cause dose-dependent oligomerization of hCSPα in neurons. WT and mutant hCSPα (L115/L116) were expressed in larval neurons of *white*[1118] animals (control) with an elav driver from one or two transgenes (2x). (A) Western blot of larval brain protein extracts probed for hCSPα. Signals for SDS-resistant hCSPα oligomers (OM), lipidated monomeric hCSP (LM), and non-lipidated hCSP (NLM) are indicated. β-tubulin was used as loading control. (B) Western blots probed for hCSPα or lysine-linked-ubiquitin of hCSPα-immunoprecipitated extracts from adult heads of indicated genotypes. Signals for IgG heavy chain are indicated. (C) Average oligomer/monomer ratios (N = 5). (D–F) Levels of lipidated (D), non-lipidated (E), and total hCSPα (F) normalized to WT hCSPα (N = 6). (G–I) Dosage-dependent increase of hCSPα oligomer (G), lipidated (H), and non-lipidated monomer levels (I). Signals were normalized to loading control and plotted as n-fold change from 1-copy expression of WT hCSPα (N = 6). (J) Levels of monomeric dCSP shown as n-fold change from control (N = 3). Graphs display mean ± SEM. Statistical analysis used one-way ANOVA (C–F, J) or two-tailed unpaired *t* test (G–I); *, p<0.05; **, p<0.01; ***, p<0.001.

DOI: https://doi.org/10.7554/eLife.46607.004

Next, we tested whether hCSPα can functionally replace endogenous dCSP and restore the premature lethality of flies lacking dCSP (*Zinsmaier et al., 1994*). Flies and mice lacking CSP due to a gene deletion exhibit progressive neurodegeneration, paralysis and premature death (*Fernández-Chacón et al., 2004*; *Zinsmaier et al., 1994*). Pan-neuronal elav-driven expression of normal hCSPα from one transgenic copy in homozygous *dcsp* deletion mutants significantly restored adult lifespan from ~4–5 days to 15–20 days (LD50 p<0.001; *Figure 1D*). In comparison, expression of fly dCSP2 also restored adult lifespan only partially (*Figure 1D*). Even though this rescue was only partial, it confirms that hCSPα is functional in flies, which made a *CLN4 Drosophila* model using human proteins tenable.

Notably, neuronal expression of hCSP-L115 and -L116 in a *dcsp* null background was able to partially rescue the lifespan deficit (p<0.001, LD50), although not nearly to the extent of WT hCSPα (*Figure 1D*). This indicates that *CLN4* mutant hCSPα proteins are at least partially functional.

## *CLN4* mutations cause formation of SDS-resistant and ubiquitinated hCSPα protein oligomers in neurons

In humans, *CLN4* mutations reduce levels of lipidated monomeric hCSPα and drive the formation of high-molecular weight, SDS-resistant hCSPα protein oligomers that are ubiquitinated (*Benitez et al., 2015*; *Greaves et al., 2012*; *Henderson et al., 2016*; *Nosková et al., 2011*). Both features were also present when mutant hCSP-L115 or -L116 were expressed in non-neuronal and neuronal mammalian cell cultures with L115 exhibiting stronger effects than L116 (*Diez-Ardanuy et al., 2017*; *Greaves et al., 2012*; *Henderson et al., 2016*; *Zhang and Chandra, 2014*).

To test whether expression of *CLN4* mutant hCSPα in *Drosophila* neurons recapitulates the pathological features of post-mortem human brains, we analyzed the properties of *CLN4* mutant hCSPα by immunoblotting, using protein extracts from larval VNCs. When expressed from a single transgene with an elav-Gal4 driver, lipidated monomeric protein levels of WT hCSPα were estimated to be 0.98 ± 0.23 times of endogenous dCSP levels (data not shown). In comparison, levels of lipidated monomeric hCSP-L115 and -L116 were significantly reduced to 13% and 40% of WT hCSPα levels, respectively (p<0.003, *Figure 2A,D*). Levels of lipidated hCSP-L115 monomers were significantly lower than hCSP-L116 (p<0.01, *Figure 2D*). Levels of non-lipidated monomeric hCSP-L115 and -L116 were comparable to WT hCSPα levels (p>0.2; *Figure 2A,E*).

Both *CLN4* mutations induced the formation of SDS-resistant, high-molecular weight hCSPα oligomers in *Drosophila* neurons (p<0.002; *Figure 2A,C*). In contrast, WT hCSPα oligomers were barely detectable (*Figure 2A,C*). The mutation L115 triggered oligomerization to a significantly larger degree than L116 (p<0.006; *Figure 2C*), which is consistent with the lower levels of hCSP-L115 monomers (*Figure 2D*). Increasing levels of DTT in the buffer to reduce disulfide bonds had little to no effect on the levels and size of hCSP-L115 and -L116 oligomers, which varied widely from ~250 kDa to more than 500 kDa (not shown).

To determine whether *CLN4* mutant hCSPα oligomers are ubiquitinated, we immunoprecipitated hCSPα from larval brain cell lysates and probed western blots with an antibody that specifically detects K29-, K48-, and K63-linked mono- and poly-ubiquitinated proteins but not free ubiquitin monomers (*Fujimuro et al., 1994*). hCSPα antibodies immunoprecipitated both hCSPα monomers and oligomers (*Figure 2B*) but not endogenous dCSP (not shown). The anti-ubiquitinated protein antibody only recognized a strong ~250 kD protein band in precipitates from L115 and L116 mutant brains, which correlated in size with high-molecular weight hCSPα oligomers (*Figure 2B*). No ubiquitin-positive signals corresponding to *CLN4* mutant hCSPα monomers, WT hCSPα monomers or oligomers were detected (*Figure 2B*). Taken together, these data suggest that expression of *CLN4* mutant hCSPα in fly brains reproduces the critical biochemical pathological features of post-mortem human *CLN4* brains (*Greaves et al., 2012*; *Henderson et al., 2016*; *Nosková et al., 2011*).

## Oligomerization of *CLN4* mutant hCSPα is dose-dependent

Oligomerization of purified *CLN4* mutant hCSPα proteins is dose-dependent in vitro (*Zhang and Chandra, 2014*). To test whether this is also the case in vivo, we doubled the gene dosage of WT and *CLN4* mutant hCSPα by expressing two copies of the respective transgenes with a single elav-Gal4 driver. In comparison to the expression from one transgene, levels of lipidated and non-lipidated monomeric WT hCSPα increased ~2.4 fold and ~3.3 fold, respectively (p<0.01, *Figure 2A,D–*

*E,H–I*). Yet, WT hCSP oligomer levels increased only 1.5-fold (p<0.005; *Figure 2G*) and remained low such that the oligomer/monomer ratio was not altered (p=0.9; *Figure 2C*).

In comparison to single copy gene expression, doubling gene dosage increased levels of lipidated hCSP-L115 monomers only modestly by 1.5-fold (p<0.04) while hCSP-L116 levels were not significantly affected (p=0.3; *Figure 2H*). However, levels of hCSP-L115 and -L116 oligomers increased 4.6- and 3.6-fold, respectively (p<0.04; *Figure 2G*). Due to the disproportional increase of *CLN4* mutant oligomers, the oligomer/monomer ratio increased from 0.9 to 2.8 for hCSP-L115 and from 0.4 to 1.8 for hCSP-L116 (*Figure 2C*). Hence, oligomerization of hCSP-L115 and -L116 is dose-dependent.

Doubling expression disproportionally increased levels of non-lipidated hCSP-L115 and -L116 monomers 8.6- and 4.6-fold, respectively (p<0.002, *Figure 2I*). The increase of non-lipidated mutant monomers is unlikely due to a rate limiting effect on mechanisms mediating palmitoylation because levels of non-lipidated WT monomers increased much less than mutant monomers, even though overall levels of lipidated WT monomers were much higher (*Figure 2E*).

Doubling gene dosage of hCSP-L115 and -L116 expression increased total overall protein levels 3.0 and 3.9-fold, respectively (p<0.002; *Figure 1—figure supplement 1D*). It also preserved the relative difference in overall protein levels between WT and *CLN4* mutants (*Figure 2F*). Since lipidated mutant monomer levels remained unaltered (p>0.1; *Figure 2D*), the increase in overall protein levels of *CLN4* mutant hCSPα is essentially due to increased levels of oligomers and non-lipidated monomers (p<0.03; *Figure 2C,E*).

Levels of endogenous dCSP monomers were not affected by low or high expression of *CLN4* mutant hCSPα (*Figure 2J*). Notably, high-molecular weight oligomers of endogenous WT dCSP were not detectable after expression of *CLN4* mutant hCSPα (*Figure 1—figure supplement 1C*), which contrasts a previous study detecting small amounts of overexpressed GFP-tagged WT hCSPα in *CLN4* mutant hCSPα oligomers (*Greaves et al., 2012*).

## Oligomerization of neuronally expressed *CLN4* mutant hCSPα precedes lethality

Elav-driven neuronal expression of WT or *CLN4* mutant hCSPα from a single transgene had no effect on viability during development (*Figure 3A*) and adult lifespan (not shown). However, doubling gene expression levels severely reduced viability during development of hCSP-L115 and -L116 animals in comparison to WT hCSPα expression (p<0.0001; *Figure 3A*). Adult L115- and L116-mutant escapers were initially sluggish in comparison to WT hCSPα but improved within ~10 days and exhibited a normal lifespan (not shown), which may be due to a decrease in the elav-driven expression. There was no significant difference in the viability between L115 and L116 mutant animals (p=0.91; *Figure 3A*). The sharp drop in viability induced by 2-copy expression of hCSP-L115 and -L116 indicates a tight threshold of neurotoxicity, which might be linked to the more than 3.6-fold increase of

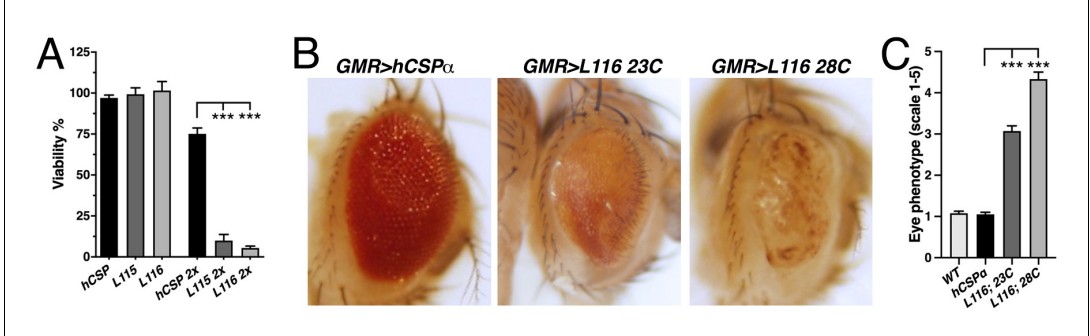

**Figure 3.** *CLN4* mutations cause dose-dependent lethality and eye degeneration. (**A**) Viability of animals expressing WT, L115, or L116 mutant hCSPα pan-neuronally from one or two transgenes (2x) with an elav driver (N ≥ 3, n ≥ 740). (**B**) Images of adult fly eyes expressing WT hCSPα or hCSPα-L116 with a GMR-Gal4 driver at 23˚C and 28˚C. (**C**) Semi-quantitative assessments of *CLN4*-induced eye phenotypes (N ≥ 9). Graphs display mean ± SEM. Statistical analysis used one-way ANOVA (**A**) and Kruskal-Wallis test (**C**); *, p<0.05; **, p<0.01; ***, p<0.001.
DOI: https://doi.org/10.7554/eLife.46607.005

*CLN4* mutant CSPα oligomers and/or the more than 4.6-fold increase of non-lipidated monomers (*Figure 2G,I*).

Expression of *CLN4* mutant hCSPα with the pan-neuronal nSyb-Gal4 driver also caused a dose-dependent lethality, which was similar to that of the elav-Gal4 driver (data not shown). In contrast to the elav-driven expression, *CLN4* mutant adult escapers were very sluggish and hardly moved. They were also unable to properly inflate their wings (*Figure 1—figure supplement 1B*) and died prematurely within 1–3 days.

To further examine potential degenerative effects in adults, we exclusively expressed hCSP-L116 in the eye by using the eye-specific GMR-Gal4 driver (*Hay et al., 1997*). High-level expression of WT hCSPα from a line with multiple transgene insertions (see Materials and methods) had essentially no effect at 23°C or 28°C in comparison to non-expressing control (p>0.9; *Figure 3B–C*). In comparison, expression of hCSP-L116 at 23°C severely impaired the size, integrity, and pigmentation of the eye (p<0.0001; *Figure 3B–C*). Raising flies at 28°C to increase Gal4 activity and thereby gene expression enhanced the severity of the degenerative phenotype of L116-mutant eyes (p<0.0001; *Figure 3B–C*). This enhancement correlates with the dose-dependent increase in hCSP-L116 oligomerization (*Figure 2C,G*) but the higher temperature may also increase misfolding and/or oligomerization of mutant hCSPα.

## *CLN4* mutations impair hCSPα's synaptic localization

Previously, it has been suggested that *CLN4* mutations may disrupt anterograde trafficking of CSPα due to either oligomer formation or impaired palmitoylation of monomers (*Benitez et al., 2011*; *Greaves et al., 2012*; *Nosková et al., 2011*). However, whether *CLN4* mutations indeed affect the synaptic localization of CSPα remained unclear. To assess this, we triple-immunostained 3rd instar larval NMJs with antibodies against hCSPα, endogenous dCSP marking SVs, and HRP marking the neuronal plasma membrane. In comparison to WT hCSPα, levels of mutant hCSP-L115 and -L116 were significantly reduced at synaptic boutons (p<0.013; *Figure 4A,H*). Expression of *CLN4* mutant or WT hCSPα had no effect on synaptic levels of endogenous WT dCSP at NMJs (p>0.6; *Figure 4A, I*). Despite the reduced synaptic levels, mutant hCSPα still partially rescued the lifespan deficit of *dcsp* deletion mutants, although not nearly to the extent of WT hCSPα (*Figure 1D*). This indicates that *CLN4* mutant hCSPα proteins are at least partially functional.

While hCSPα co-localized with endogenous dCSP uniformly in the periphery of synaptic boutons (*Figure 4A,C*), hCSP-L115 and -L116 were enriched in brightly stained clusters that were more distant from the presynaptic membrane (*Figure 4A,C*). In comparison to control, *CLN4* mutant hCSPα also accumulated abnormally in axons of segmental nerves (*Figure 4G*) and the larval brain (p<0.009; *Figure 4B,J*; *Figure 4—figure supplement 1A*). The size of *CLN4* mutant hCSPα accumulations in the larval brain was heterogeneous, ranging from ~500 nm in diameter to ~3 μm. Occasionally, extreme accumulations of mutant hCSPα were observed at synaptic boutons of larval NMJs (*Figure 4—figure supplement 1C*). The occurrence of abnormal hCSP-L115 and -L116 accumulations was dose-dependent (not shown), which was most pronounced in axons of segmental nerves (*Figure 4G*). Hence, *CLN4* mutations reduce synaptic levels of hCSPα and cause a severe mislocalization.

## Abnormal accumulations of *CLN4* mutant hCSPα contain WT dCSP and ubiquitinated proteins

A fraction of the abnormal accumulations of hCSP-L115 and -L116 in axons (not shown) and somata (*Figure 4E* and *Figure 4—figure supplement 1A*) contained significant amounts of endogenous WT dCSP. Since WT dCSP was not detected together with high-molecular weight oligomers of *CLN4* mutant hCSPα on Western blots (*Figure 1—figure supplement 1C*), this indicates that *CLN4* mutant hCSPα and WT dCSP may co-accumulate on intracellular membranes.

Theoretically, *CLN4* mutant hCSPα could accumulate on ER, Golgi, or abnormal SVP membranes during secretory trafficking. Alternatively, mutant hCSPα could accumulate on endosomal membranes that are targeted for degradation. To test whether mutant hCSPα accumulations contain ubiquitinated proteins indicative of prelysosomal membranes, we used antibodies that exclusively detect ubiquitinated proteins but not monomeric ubiquitin (*Fujimuro et al., 1994*). Indeed, essentially all *CLN4* mutant hCSPα accumulations were immunopositive for ubiquitinated proteins in

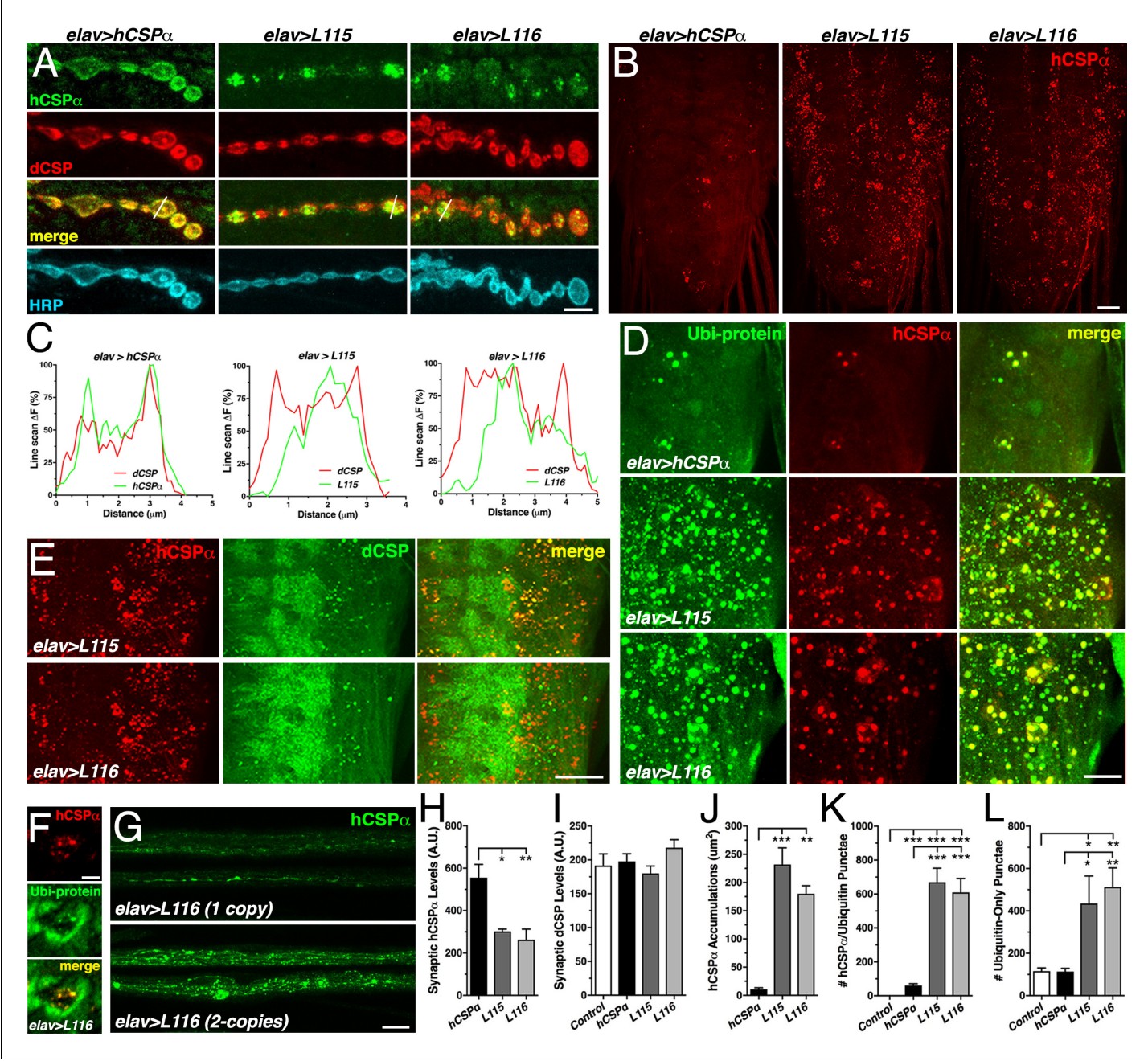

**Figure 4.** *CLN4* mutations reduce synaptic hCSPα levels and cause abnormal accumulations with endogenous dCSP and ubiquitinated proteins in axons and somata. WT, L115 or L116 mutant hCSPα were expressed in larval neurons with an elav-Gal4 driver from one (**D–E, J–L**) or two (**A, H–I**) transgenes. Genotypes are indicated. (**A**) Larval NMJs immunostained for hCSPα, endogenous dCSP, and the neuronal membrane marker HRP. White lines denote line scans shown in C. (**B**) Larval VNCs stained for hCSPα. (**C**) Plots of hCSPα and dCSP fluorescence from single line scans through synaptic boutons (white lines in A). (**D**) Larval VNC segments stained for hCSPα (red) and lysine-linked-ubiquitin visualizing ubiquitinated proteins (Ubi-protein). (**E**) Larval brain segments stained for hCSPα and dCSP. (**F**) Synaptic bouton of larval NMJ stained for hCSPα and Ubi-proteins. (**G**) Proximal larval segmental nerves stained for hCSPα. (**H–I**) Average levels of hCSPα (**H**) and dCSP (**I**) at synaptic boutons of larval NMJs (N > 4). (**J**) Cumulative area of abnormal hCSPα accumulations in larval brains (N ≥ 3). (**K–L**) Average number of accumulations immunopositive for both hCSPα and ubiquitin (**K**), or only positive for ubiquitin (**L**) but not hCSPα (N ≥ 4). Scale bars: 5 µm (**A**), 20 µm (**B, G**), 15 µm (**D**), 10 µm (**E**), 5 µm (**F**). Graphs display mean ± SEM. Statistical analysis used one-way ANOVA (**H–L**); *, $p < 0.05$; **, $p < 0.01$; ***, $p < 0.001$.

DOI: https://doi.org/10.7554/eLife.46607.006

The following figure supplements are available for figure 4:

**Figure supplement 1.** hCSP-L115 and -L116 abnormally accumulate with endogenous dCSP and ubiquitinated proteins.

*Figure 4 continued on next page*

Figure 4 continued

DOI: https://doi.org/10.7554/eLife.46607.007

**Figure supplement 2.** Effects of *CLN4*-analogous mutations in dCSP.

DOI: https://doi.org/10.7554/eLife.46607.008

somata and synaptic boutons of NMJs (*Figure 4D,F*, and *Figure 4—figure supplement 1B*). In comparison to larvae expressing WT hCSPα, the amount of hCSPα accumulations containing ubiquitinated proteins was significantly increased in *CLN4* mutants (p<0.001; *Figure 4K*). Hence, the abnormal accumulations of *CLN4* mutant hCSPα are either enriched in ubiquitinated hCSPα oligomers and/or contain other ubiquitinated proteins that are potentially destined for protein degradation.

The co-immunostainings against hCSPα and ubiquitinated proteins also revealed a substantial increase in the levels of ubiquitinated proteins that was not associated with accumulations of *CLN4* mutant hCSPα (*Figure 4D* and *Figure 4—figure supplement 1B*). Brains of control (*w^1118^*) and larvae expressing WT hCSPα contained a similar amount of ubiquitinated protein foci that were negative for hCSPα (*Figure 4L*), indicating that expression of WT hCSPα exerts little to no effect on protein ubiquitination. In comparison, the amount of ubiquitinated protein foci that were not associated with mutant hCSPα was significantly increased in brains expressing hCSPα-L115 or -L116 (p<0.03; *Figure 4L*). Hence, *CLN4* mutant hCSPα may directly or indirectly affect a step of protein homeostasis that that leads to excessive protein ubiquitination of unrelated proteins.

## *CLN4* mutations in fly and human CSP have similar effects

To validate the observed phenotypes of *CLN4* mutant hCSPα and exclude that they are not an artifact of expressing mutant human proteins in fly neurons, we expressed *CLN4* mutant dCSP2 (dCSP2) containing the mutations V117R and I118Δ, which are analogous to the mutations L115R and L116Δ in hCSPα (*Figure 1A*). All fly transgenes were inserted at the same transgenic insertion site as the human transgenes. Elav-driven neuronal expression of dCSP2, dCSP-V117, or -I118Δ from a single transgene had no effect on viability during development at 23°C (not shown) and 27°C (p>0.4; *Figure 4—figure supplement 2A*). Adult lifespan at 27°C was also normal (*Figure 4—figure supplement 2D*). However, doubling gene expression levels severely reduced the viability of animals expressing WT dCSP, dCSP-V117 or -I118Δ (*Figure 4—figure supplement 2B*). Notably, the few adult WT dCSP and dCSP-V117 animals exhibited rough eyes, abnormally inflated wings, impaired locomotion and died within a day (not shown). Adult flies expressing dCSP-I118 were never observed.

Like for *CLN4* mutant hCSPα, levels of monomeric lipidated dCSP-V117 and -I118 were reduced in comparison to WT dCSP (*Figure 4—figure supplement 2C*). Similar to its human analog L115 (*Figure 2A*), V117 reduced levels of monomeric dCSP more severely (*Figure 4—figure supplement 2C*). In addition, both V117 and I118 triggered the formation of SDS-resistant, high-molecular weight oligomers (*Figure 4—figure supplement 2C*), which were absent for WT dCSP2 expression. Expressing dCSP2-V117 and -I118 in *dcsp* deletion mutants had similar effects (*Figure 4—figure supplement 2C*). Hence, the pathological features of the dominant *CLN4* mutations reducing levels of lipidated CSP monomers and triggering protein oligomerization are conserved between fly and human CSP.

To determine the subcellular localization of *CLN4* mutant dCSP in fly neurons, we expressed normal and *CLN4* mutant proteins from a single transgene in neurons of homozygous *dcsp* deletion null mutants. Like for mutant hCSPα, levels of dCSP-V117 and -I118 were reduced at synaptic boutons of larval NMJs and the neuropil of the larval VNC (*Figure 4—figure supplement 2E–F*). Mutant dCSPs also accumulated abnormally in axons and neuronal somata of the larval brains (*Figure 4—figure supplement 2E*).

## *CLN4* mutant hCSPα accumulates on prelysosomal endosomes

To further define the nature of the co-accumulations of *CLN4* mutant hCSPα with ubiquitinated proteins, we tested a number of organelle markers for a potential colocalization. Mutant hCSPα

accumulations did not colocalize with the endoplasmic reticulum (ER), cis-, or trans-Golgi complexes (*Figure 5A*), which excluded a major defect in ER or Golgi trafficking.

A large fraction of mutant hCSPα co-accumulated with hLAMP1-GFP (*Figure 5B–C*), which labels a heterogeneous population of organelles ranging from pre-degradative endosomal species to degradative lysosome (*Cheng et al., 2018*; *Saftig and Klumperman, 2009*; *Yap et al., 2018*). A similar large fraction of mutant hCSPα co-accumulated with hepatocyte growth factor regulated tyrosine kinase substrate (HRS, *Figure 5C,E*), which is a critical component of 'endosomal sorting complexes required for transport' (ESCRT) mediating the transition from early to late endosomes (*Raiborg and Stenmark, 2009*). Only a small fraction co-localized with coexpressed Rab5-GFP (not shown) or the autophagosomal marker ATG8/LC3-GFP (*Figure 5D*). No co-localization was observed with lysosomal Spinster-GFP (*Figure 5B*; *Rong et al., 2011*; *Sweeney and Davis, 2002*) or the late endosomal protein Rab7 (not shown; *Guerra and Bucci, 2016*).

Notably, endosomes accumulating mutant hCSPα did not co-localize with the co-expressed phosphatidylinositol 3-phosphate (PI$_3$P) sensor FYVE-GFP (*Figure 5B*), even though HRS requires PI$_3$P for membrane association (*Mayers et al., 2013*; *Raiborg et al., 2001a*). Hence, this suggests that all

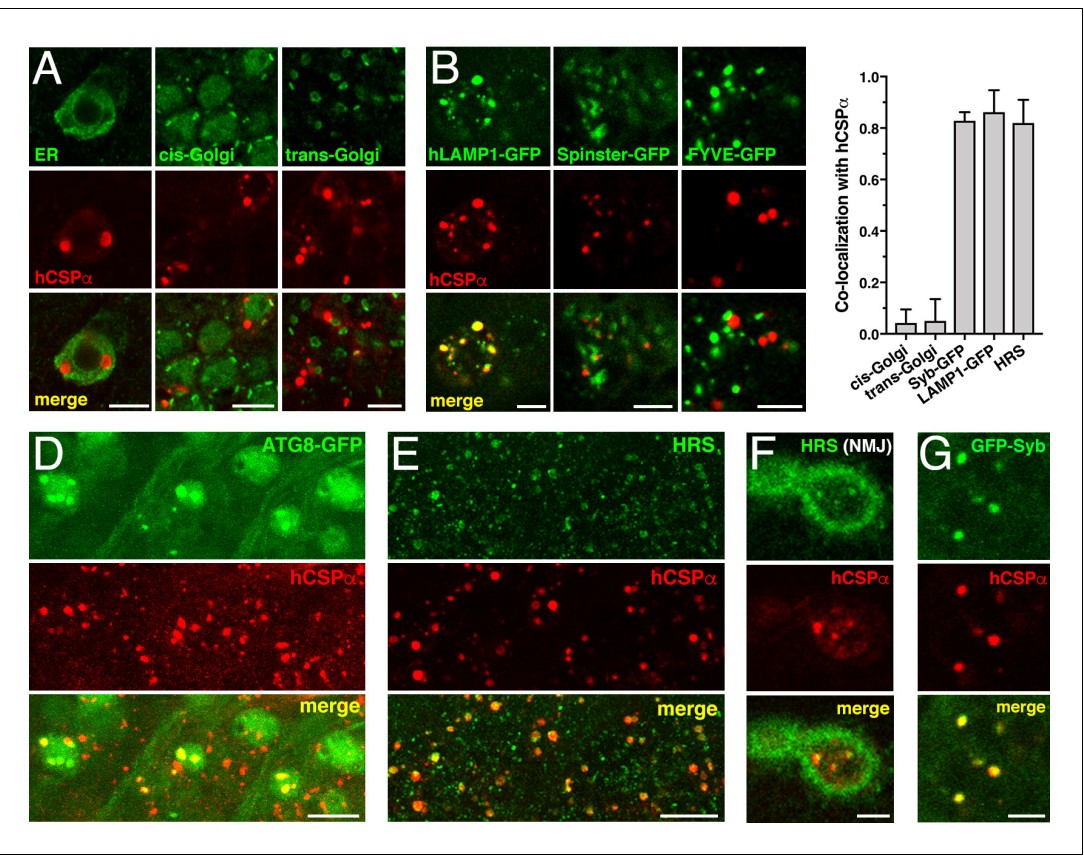

**Figure 5.** *CLN4* mutant hCSPα accumulates on LAMP1- and HRS-positive endosomes. hCSP-L116 was expressed in larval neurons with an elav-Gal4 driver from one transgene. As indicated, respective reporter transgenes were co-expressed. (**A**) Neurons of larval VNCs co-immunostained for hCSPα and the ER marker GFP-KDEL, the cis-Golgi marker GMAP, or the trans-Golgi marker Golgin 245. (**B**) Neurons co-immunostained for hCSPα (red) and co-expressed hLAMP1-GFP, Spinster-GFP or the PI$_3$P marker FYVE-GFP. (**C**) Fraction of organelle markers colocalizing with hCSPα accumulations (mean ± SEM; n ≥ 65, N ≥ 4). (**D–E**) Segments of larval brains costained for hCSPα and ATG8/LC3-GFP (**D**) or HRS (**E**). (**F**) Synaptic boutons at larval NMJs co-stained for hCSPα and HRS. (**G**) Neuron co-immunostained for hCSPα and coexpressed GFP-nSyb. Scale bars: 10 μm (**D–E**), 5 μm (**A–B, F–G**).
DOI: https://doi.org/10.7554/eLife.46607.009

The following figure supplement is available for figure 5:

**Figure supplement 1.** RNAi-mediated KD of TSG101 causes accumulation of dCSP on HRS-positive endosomes.
DOI: https://doi.org/10.7554/eLife.46607.010

PI$_3$P binding sites may be occupied on the abnormal endosomes accumulating mutant hCSPα. Alternatively, PI$_3$P may be absent from these endosomes, which would imply an abnormal retention of HRS. Taken together, these data suggest that *CLN4* mutant hCSPα accumulates on prelysosomal endosomes that are inefficiently processed for lysosomal fusion.

Consistent with the colocalization of mutant hCSPα with ubiquitinated proteins at axon terminals (*Figure 4F*), HRS also colocalized with mutant hCSPα at synaptic boutons of NMJs (*Figure 5F*). This raised the possibility that hCSPα-positive endosomes may originate from synaptic terminals. Consistently, hCSPα-positive accumulations co-localized with SV-associated Synaptobrevin-GFP (Syb-GFP; *Figure 5C,G*) while a small subset co-localized with the synaptic plasma membrane protein Syntaxin 1A (not shown). The CSPα chaperone clients Dynamin and SNAP-25 (*Sharma et al., 2011*; *Zhang et al., 2012*) did not co-localize with hCSPα-positive endosomes (not shown).

## *CLN4* mutations cause various ultrastructural endo-membrane abnormalities in axons and neuronal somata

Since *CLN4* mutant hCSPα abnormally accumulates on prelysosomal endosomes, we used electron microscopy to detect potential ultrastructural defects in membrane trafficking. Expression of hCSPα-L115 and -L116 induced highly abnormal membrane structures in neuronal somata, the neuropil of the larval VNC, and axons of segmental nerves (*Figure 6*). The most prominent and frequent abnormal structures were multilamellar 'membrane whirls' of various shapes and density that contain highly electron-dense membranes (arrowheads, *Figure 6A–B,C–D,F–G*). Like endosomal hCSPα-positive accumulations detected by confocal microscopy, membrane whirls were most frequently observed in the neuropil of the VNC and axons of segmental nerves (*Figure 6C–D*). To a lesser degree, they were present in the cytoplasm of neuronal somata (*Figure 6A–B,F–G*) and occasionally the nucleoplasm (not shown). Sporadically, whirls were found on opposite sides of plasma membranes of neighboring cells (arrowhead, *Figure 6B*).

In addition to electron-dense membrane whirls, a number of secondary (residual) lysosomes (*Figure 6A,E*) and abnormal autophagosome- and/or amphisome-like structures were present (white arrowhead, *Figure 6F–I*). Interestingly, EM-dense membranes forming whirls may interact with autophagosomes (arrow, *Figure 6G*). Notably, somewhat similar 'membrane whirl' and autophagosome-like structures were observed in various ESCRT loss of function mutants including TSG101, Snf7/CHMP4B, VPS4 and CHMP2B (*Doyotte et al., 2005*; *Lee et al., 2007*; *Razi and Futter, 2006*).

Next to abnormal membranes structures, homogenous electron-dense accumulations of unknown nature were frequently present in cellular regions of *CLN4* mutant larval brains (arrow, *Figure 6B,H*). A limiting membrane was not detectable for these EM-dense accumulations, which are reminiscent but not identical of granular osmophilic deposits (GRODs) seen in post-mortem human tissue of *CLN4* patients (*Anderson et al., 2013*; *Burneo et al., 2003*; *Nosková et al., 2011*; *Virmani et al., 2005*). Since these structures were absent from the VNC neuropil and segmental nerves, they are unlikely a correlate for the abnormal endosomal accumulations of mutant hCSPα in axons.

Finally, a number of neurons that contained abnormal membrane structures also exhibited a severe fragmentation of the nuclear envelope and bloated Golgi cisternae (*Figure 6B*), which together likely indicate a late stage of neuronal dysfunction and neurodegeneration (*Nixon, 2006*).

## Loss of ESCRT function causes endosomal accumulations of endogenous dCSP but no oligomerization

To verify that WT CSP is normally trafficked through the endo-lysosomal pathway, we impaired ESCRT function by an RNAi-mediated knock down (KD) of tumor suppressor gene 101 (TSG101), which is part of the ESCRT-I complex acting downstream of HRS and required for late endosome formation (*Doyotte et al., 2005*; *Razi and Futter, 2006*). Neuronal KD of TSG101 caused a mislocalization of endogenous dCSP, which accumulated on HRS-positive endosomes in the neuropil and neuronal somata of the larval VNC (*Figure 5—figure supplement 1A*). Abnormal dCSP accumulations were also present in central regions of synaptic boutons of larval NMJs (*Figure 5—figure supplement 1B*). Hence, WT CSP is likely being degraded though the endo-lysosomal pathway. This conclusion is consistent with the low amounts of lysosome-associated CSPα in human fibroblasts, N2A cells, mouse neurons, and lysosome-enriched fractions (*Benitez and Sands, 2017*; *Chapel et al., 2013*; *Schröder et al., 2007*; *Tharkeshwar et al., 2017*).

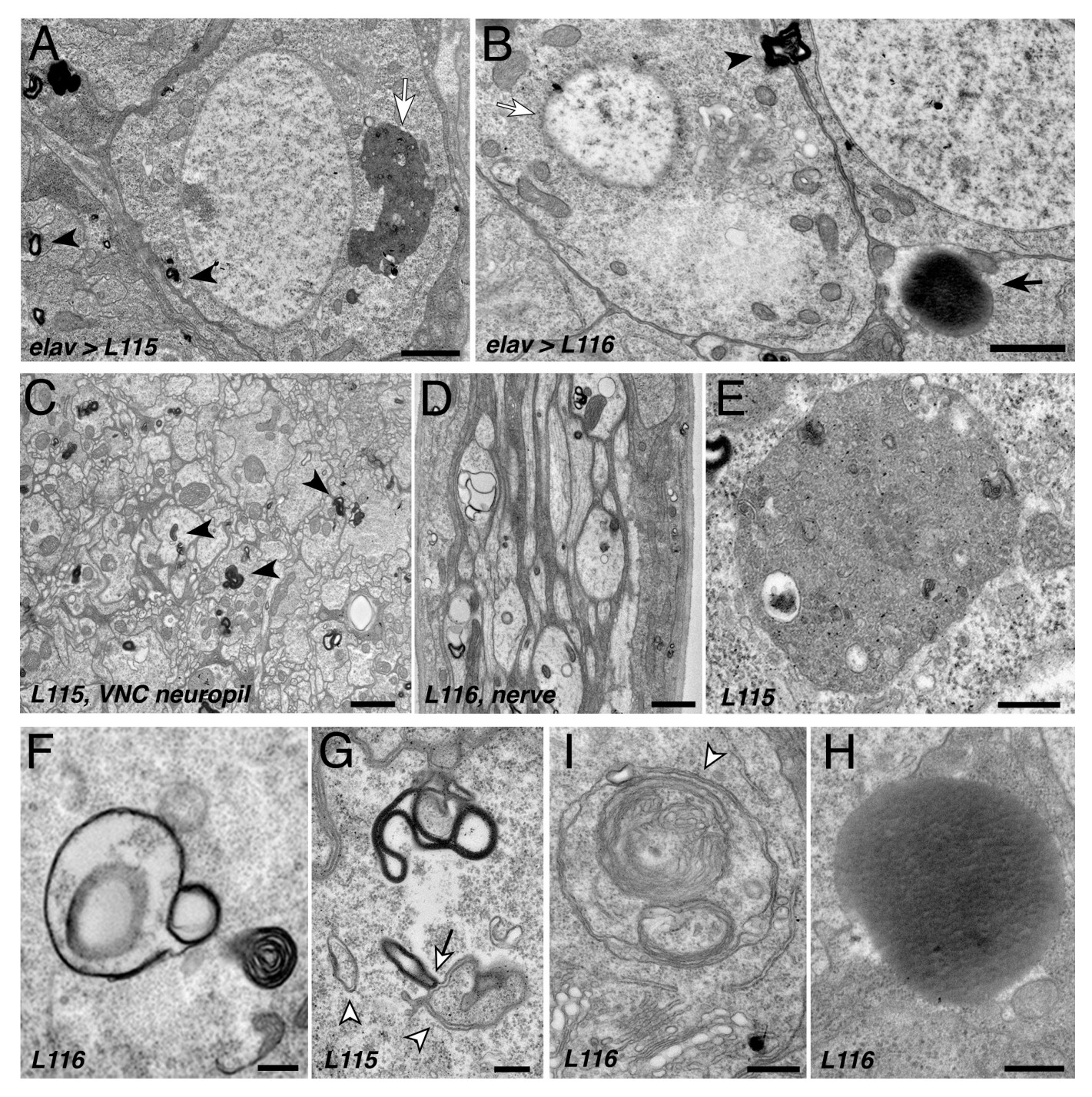

**Figure 6.** *CLN4* mutations cause abnormal endomembrane structures and EM-dense accumulations. TEM micrographs of ultrathin (70 nm) sections from larval VNCs expressing hCSP-L115 or -L116 with an elav driver from two transgenic copies. Genotypes are indicated. (A–B) Neuronal somata containing electron-dense membrane whirls (black arrowheads), large electron-dense extracellular deposits (black arrow, (B), and occasionally 'residual lysosomes' (white arrow, (A), bloated Golgi Apparati (B), and degenerating nuclear membranes (white arrow, (B). (C) Neuropil of larval VNC containing membrane whirls in neuronal processes (black arrowheads). (D) Sagittal section of larval segmental nerve containing membrane whirls and abnormal autophagosome-like structures in axons of sensory and motor neurons. (E–H) High magnification images showing residual lysosome with a diverse variety of intraluminal vesicles (E), various forms of EM-dense membrane whirls (F–G) and autophagosome-like structures (white arrowheads, (G–I) that may interact with EM-dense structures (white arrow, (G), and an electron-dense extracellular deposit (H). Scale bars: 1 μm (A–C), 200 nm (D–E).
DOI: https://doi.org/10.7554/eLife.46607.011

Notably, KD of TSG101 essentially mimicked the abnormal endosomal accumulation of *CLN4* mutant hCSPα-L115 and -L116 with one important exception. In contrast to hCSPα-L115 and -L116 expression, KD of TSG101 did not cause the formation of high-molecular weight dCSP oligomers (*Figure 5—figure supplement 1C*), indicating that CSP oligomer formation is independent of impaired ESCRT function. In contrast to a recent report showing that impaired lysosomal activity in mouse neurons decreases CSPα's palmitoylation (*Sambri et al., 2017*), impaired prelysosomal ESCRT function had no effect on the palmitoylation state of dCSP.

## The *CLN4* alleles hCSP-L115 and -L116 are hypermorphic gain of function mutations

Several hypotheses regarding the genetic nature of the dominant *CLN4* mutations have been suggested. Mutant hCSPα monomers and/or oligomers could act as dominant-negatives that progressively sequester WT hCSPα and deplete the functional hCSPα pool, which in turn could trigger neurodegeneration (*Greaves et al., 2012*; *Nosková et al., 2011*; *Zhang and Chandra, 2014*). Alternatively, *CLN4* mutant alleles may induce toxicity via a different gain of function (GOF) mechanism (*Greaves et al., 2012*; *Henderson et al., 2016*; *Zhang et al., 2012*). Such a GOF mutation could increase a normal activity of the protein (hypermorphic mutation) or introduce a new activity (neomorphic mutation). In theory, either type of mutation could trigger neurotoxicity.

To genetically address the genetic nature of *CLN4* mutations, we examined to what degree alterations of WT dCSP or hCSPα levels may affect L115 and L116 phenotypes. The rationale behind these genetic experiments is simple: lowering WT levels should enhance dominant-negative phenotypes, reduce hypermorphic phenotypes due to an increased normal activity, or have no effect on neomorphic phenotypes due to a new protein activity (*Muller, 1932*; *Wilkie, 1994*).

Reducing endogenous WT dCSP levels by expressing two transgenic copies of *CLN4* mutant hCSP in heterozygous *dcsp* deletion mutants significantly suppressed the lethality induced by *CLN4* mutations such that significantly more animals reached adulthood (p<0.02, *Figure 7A*). Conversely, increasing levels of WT hCSPα by co-expressing WT hCSPα with either one copy of hCSP-L115 or -L116 significantly reduced viability to ~54% and 48%, respectively (p<0.001, *Figure 7B–C*). The lethality induced by co-expression of WT and *CLN4* mutant hCSPα was significantly higher than the lethality induced by two copy expression of WT hCSPα (p<0.03, *Figure 7B–C*). Hence, the modulating effects of altered WT dCSP or hCSPα levels on L115- and L116-induced lethality essentially exclude the possibility that either mutation acts as a dominant-negative. Instead, these effects are consistent with the genetic characteristics of a hypermorphic GOF mutation (*Muller, 1932*; *Wilkie, 1994*).

Next, we determined whether altered WT CSP levels may modulate the tendency of *CLN4* mutant hCSPα to form abnormal high-molecular weight oligomers. Reducing WT dCSP levels by expressing *CLN4* mutant hCSPα in hetero- or homozygous *dcsp* deletion mutants significantly attenuated the levels of SDS-resistant hCSP-L116 and -L115 oligomers (p<0.05; *Figure 7D,F*). Levels of hCSP-L116 and -L115 oligomers were attenuated to a similar degree (p>0.5; *Figure 7F*). Conversely, increasing WT dCSP levels by co-expressing a WT transgene together with a single *CLN4* mutant transgene increased the amount of hCSP-L115 and L116 oligomers (p<0.007, *Figure 7D,I*). Coexpression of WT hCSPα also significantly increased oligomerization of both hCSP-L116 and -L115 (p<0.008; *Figure 7E,L*). In comparison, high-level overexpression of WT hCSPα from two transgenes did not induce significant oligomerization (*Figures 7E* and *2A,C*).

In contrast to the modulatory effects of WT CSP on mutant hCSPα oligomer levels, reducing or even abolishing WT dCSP levels had no effect on the levels of lipidated monomeric hCSP-L115 and -L116 (p>0.4; *Figure 7G*). Coexpressing WT dCSP2 or hCSPα with *CLN4* mutant hCSPα increased lipidated mutant hCSPα monomers (p<0.05; *Figure 7J,M*). Notably, coexpression of WT hCSPα increased lipidated hCSP-L115 and -L116 monomers to levels that were similar to those of two copy WT hCSPα expression (*Figure 7M*). This indicates that WT hCSPα may stabilize *CLN4* mutant hCSPα monomers.

Levels of non-lipidated monomeric *CLN4* mutant hCSP were unaffected by reducing the gene dosage of endogenous WT dCSP2 (p>0.3; *Figure 7H*). However, coexpression of either WT dCSP2 or hCSPα with *CLN4* mutant hCSPα increased levels of non-lipidated mutant hCSPα monomers (p<0.05; *Figure 7K,N*). At least the increase induced by WT hCSPα was non-additive since it was significantly larger than the increase induced by doubling WT hCSPα expression (p<0.013; *Figure 7N*).

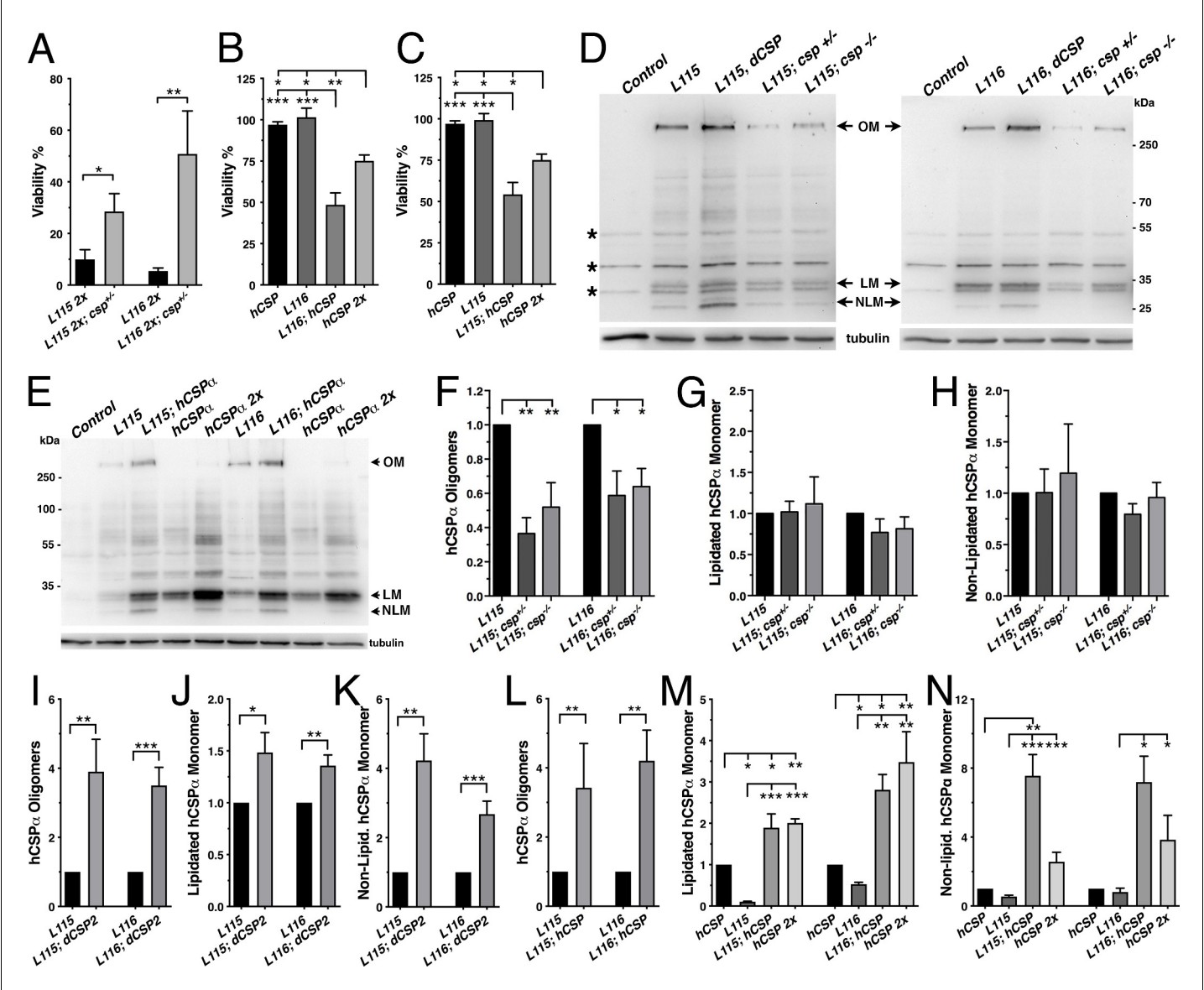

**Figure 7.** Altering wild type CSP levels modifies *CLN4* phenotypes. WT hCSPα, hCSP-L115 or -L116 were expressed in neurons from one or two transgenes (2x) with an elav driver in control (*w^1118^*), heterozygous *csp^X1/+^*, and homozygous *csp^X1/R1^* deletion mutants, or co-expressed with WT hCSPα or dCSP. Genotypes are indicated. (A–C) Effects of reducing endogenous dCSP (A) or co-expressing WT hCSPα (B–C) on the viability of hCSP-L115 and -L116 mutant flies (N > 3; n > 144). (D–E) Immunoblots of protein extracts from larval VNC of indicated genotype probed for hCSPα and β-tubulin (loading control). hCSPα oligomers (OM), lipidated (LM), non-lipidated hCSPα monomers (NLM), and unspecific signals (*) are denoted. (F–N) Effects of reduced endogenous dCSP (F–H), increased dCSP (I–K), and increased WT hCSPα (L–N) levels on hCSPα oligomers (F, I, L), lipidated monomers (G, J, M), and non-lipidated monomers (H, K, N). Signals were normalized to loading control and plotted as n-fold change of L115 or L116 levels when expressed in a WT background (N = 5). Graphs display mean ± SEM. Statistical analysis used unpaired *t* test (A, I–L) or one-way ANOVA (B–C, F–H, M–N); *, p<0.05; **, p<0.01; ***, p<0.001.

DOI: https://doi.org/10.7554/eLife.46607.012

Hence, increased levels of WT CSP may outcompete *CLN4* mutant hCSPα for palmitoylation or, alternatively, promote its depalmitoylation.

Finally, we tested whether altering levels of endogenous dCSP also affects other phenotypes of the *CLN4* fly model. Indeed, reducing dCSP levels by one gene copy significantly suppressed the endosomal accumulation of both hCSP-L115 and -L116 in larval VNCs (p<0.05; *Figure 8A–B*). Reducing dCSP gene dosage also decreased the amount of accumulating ubiquitinated proteins in the

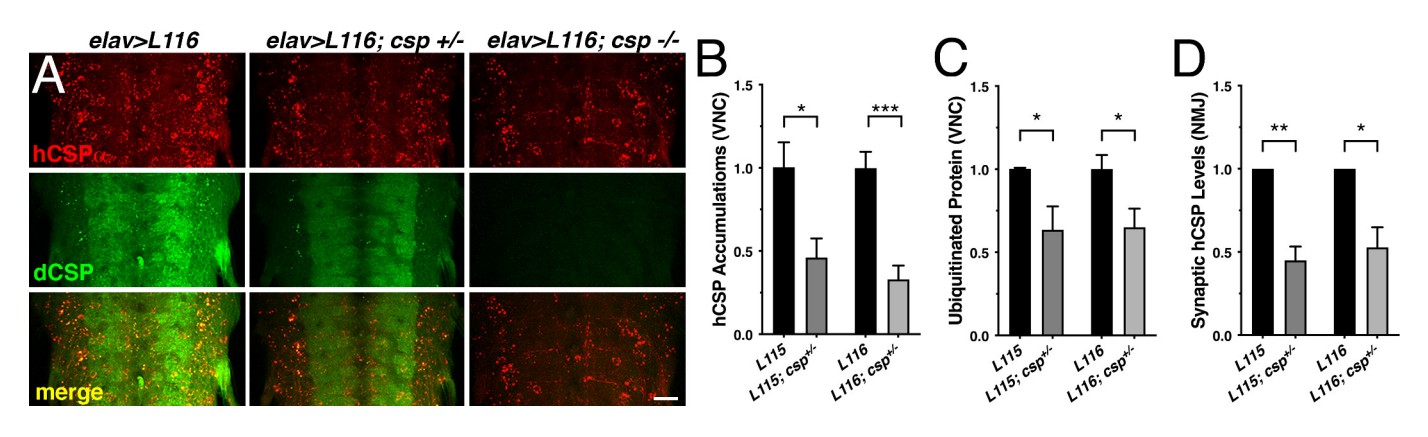

**Figure 8.** Effects of reduced dCSP levels on *CLN4* mutant hCSPα localization and protein ubiquitination. WT hCSPα, hCSP-L115 or -L116 were expressed in neurons with an elav driver in control (*w$^{1118}$*), heterozygous *csp$^{X1/+}$*, or homozygous *csp$^{X1/R1}$* deletion mutants. (A) Larval VNC stained for hCSPα and dCSP. Genotypes are indicated. Scale bar, 20 μm. (B–D) Effects of reduced *dcsp* gene dosage on the accumulation of hCSP-L115 and -L116 in larval VNC (B, N ≥ 3), ubiquitinated protein levels in larval VNCs (C, N ≥ 4), and synaptic levels of hCSP-L115 and -L116 at larval NMJs (D, N ≥ 4). Signals were normalized and plotted as n-fold change to levels of elav-L115 and -L116 expression in WT control background. Graphs display mean ± SEM. Statistical analysis used a paired (D) and unpaired *t* test (B–C); *, p<0.05; **, p<0.01; ***, p<0.001.
DOI: https://doi.org/10.7554/eLife.46607.013

larval VNC (p<0.04; *Figure 8C*). However, reduced dCSP levels significantly enhanced the already reduced synaptic protein levels of hCSP-L115 and -L116 at larval NMJs (p<0.03; *Figure 8D*). The underlying cause for the latter effect is unclear.

Taken together, the modulating effects of altering WT fly and human CSP levels on L115- and L116-induced phenotypes in viability, mutant hCSPα oligomerization, accumulation on endosomes and ubiquitinated proteins suggest that the hCSP-L115 and -L116 alleles are hypermorphic gain of function mutations that increase an intrinsic activity. This conclusion is consistent with a recent study showing similar gene dosage effects of WT CSPα on hCSP-L115R phenotypes in mouse CSPα-deficient fibroblasts (*Benitez and Sands, 2017*). In addition, these findings uncover a correlation between the reduced viability of *CLN4* mutant flies, the degree of mutant hCSPα oligomerization and the failure of degrading oligomers.

### Partial loss of CSPα's chaperone partner Hsc70-4 suppresses *CLN4*-induced phenotypes

The hypermorphic nature of *CLN4* mutations raised the possibility that at least some of the induced phenotypes are caused by an abnormally increased co-chaperone activity of CSP with the molecular chaperone Hsc70, which normally ensures efficient SV exo- and endocytosis by likely chaperoning SNARE proteins and Dynamin (*Chandra et al., 2005*; *Nie et al., 1999*; *Sharma et al., 2012*; *Sharma et al., 2011*; *Zhang et al., 2012*). To address this possibility, we tested whether altered levels of SNARE proteins or Hsc70 may modify the eye phenotypes induced by GMR-driven expression of *CLN4* mutant hCSPα (*Figure 9A*).

Reducing endogenous Hsc70-4 (Hsc4) levels by expressing hCSP-L116 in heterozygous *hsc4$^{Δ356}$* deletion mutants (*Bronk et al., 2001*) significantly suppressed both the structural and depigmentation defects of L116 mutant eyes (p=0.005; *Figure 9A,E*). Co-expression of a UAS hairpin transgene knocking down Hsc4 also suppressed the L116 eye phenotype (p=0.007; *Figure 9A,E*). However, reducing the gene dosage of Hsc70-3 or Hsc70-5 had no effect (not shown), indicating that the suppression of L116-mutant eye phenotypes by reduced levels of Hsc4 is not a general effect of Hsc70 proteins. Co-expression of either Syntaxin 1A (Syx1A) or neuronal Synaptobrevin (nSyb) with hCSP-L116 enhanced the L116 eye phenotype (p<0.01; *Figure 9A,D*). Overexpression or RNAi-mediated knockdown of SNAP25 had no effect (not shown), even though a heterozygous deletion of SNAP25 enhanced degenerative phenotypes of hCSPα knockout mice (*Sharma et al., 2012*). Individual

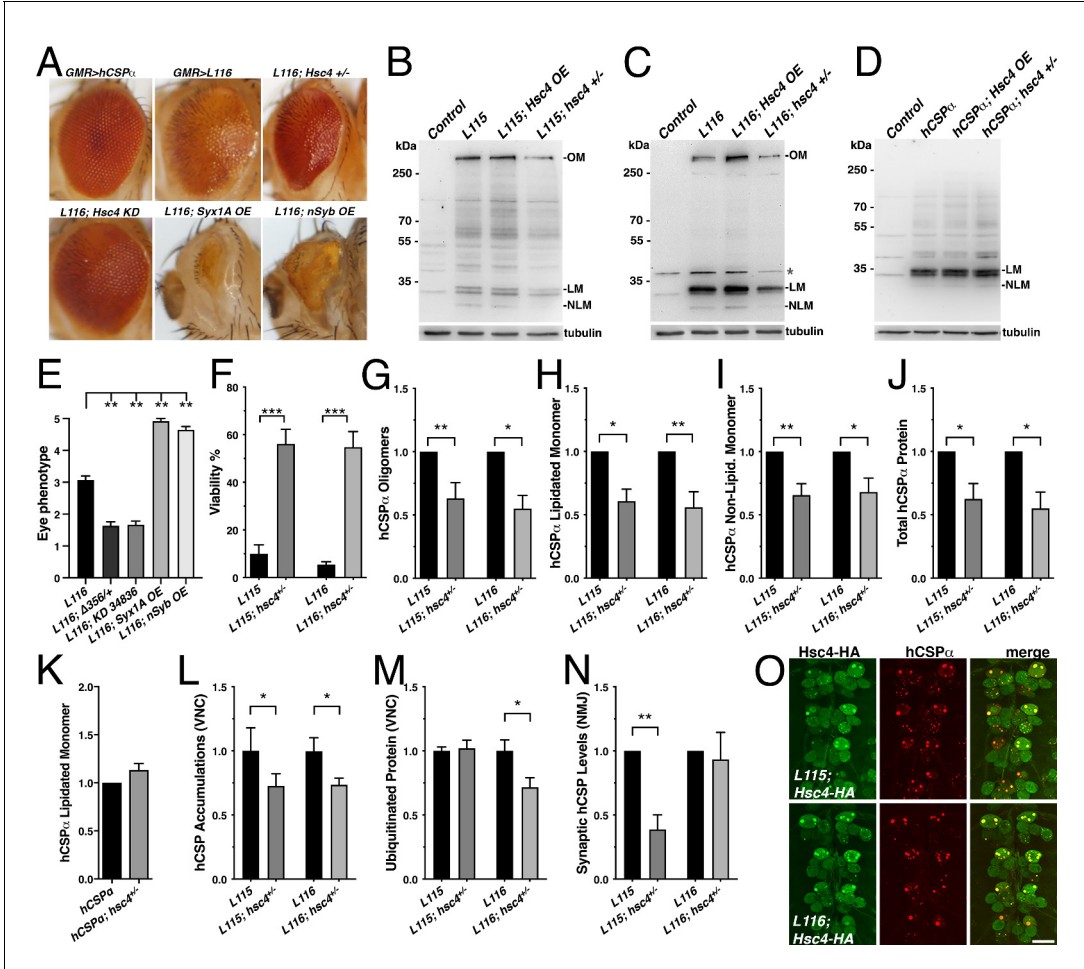

**Figure 9.** Reducing the gene dosage of Hsc4 attenuates *CLN4* phenotypes. (**A**) Adult eyes of flies expressing WT hCSPα (control) or hCSP-L116 with a GMR-Gal4 driver in the absence or presence of a heterozygous Hsc4+/- (*hsc4^Δ356*) deletion, a Hsc4 KD (34836), OE of Syx1A or Syb. (**B–D**) Immunoblots of extracts from larval VNCs of indicated genotypes were probed for hCSPα. Lipidated monomeric hCSP (LM), non-lipidated hCSP (NLM) and hCSPα oligomers (OM) are indicated. Respective transgenes were expressed with an elav-driver; control was *w^1118*. β-tubulin was used as loading control.( **E**) Semi-quantitative assessments of genetic modifier effects on *L116*-induced eye phenotypes (N ≥ 12).( **F**) Viability of animals expressing hCSPα-L115 or -L116 from two transgenes driven by elav-Gal4 in a control (*w^1118*) or a heterozygous *hsc4^Δ356* deletion background (N ≥ 3; n > 74). (**G-K**) Effects of reduced *hsc4* gene dosage on levels of hCSPα oligomers (**G**), lipidated monomers (**H**), non-lipidated monomers (**I**) and total protein levels (**J**). Signals were normalized to loading control and plotted as n-fold change to levels induced by elav-driven expression of hCSP-L115 or -L116 in a WT control background (N > 6). (**K**) Effect of reduced *hsc4* gene dosage on lipidated WT hCSPα monomer levels (N = 4). (**L**) Effects of reduced *hsc4* gene dosage on endosomal accumulations of hCSP-L115/L116 in larval VNC (N ≥ 4). (**M–N**) Effects of reduced *hsc4* gene dosage on ubiquitinated protein levels in larval VNCs (M, N = 5) and synaptic hCSP-L115/L116 expression levels at larval NMJs (N, N = 6). Signals were normalized and plotted as n-fold change from levels of elav-L115/L116 expression in WT control background. (**O**) Motor neuron somata of larval VNCs co-expressing HA-tagged Hsc4 with mutant hCSPα-L115 or -L116 stained for HA and hCSPα. Scale bars: 20 μm (K), 15 μm (M). Graphs display mean ± SEM. Statistical analysis used an unpaired *t* test (**F, M**), paired *t* test (**L, N**), or one-way ANOVA (**E, G–J**); *, p<0.05; **, p<0.01; ***, p<0.001.
DOI: https://doi.org/10.7554/eLife.46607.014

The following figure supplement is available for figure 9:

**Figure supplement 1.** Effects of altered *hsc4* gene dosage on *CLN4* mutant hCSPα protein expression and localization.
DOI: https://doi.org/10.7554/eLife.46607.015

overexpression of Syntaxin 1A (Syx1A), neuronal Synaptobrevin (nSyb), or SNAP25 had no effect on the eye (not shown).

To further test whether Hsc4 may at least in part contribute to *CLN4* phenotypes, we examined effects of altered Hsc4 levels on the lethality induced by pan-neuronal expression of *CLN4* mutant hCSPα. Reducing the gene dosage of *hsc4* by expressing mutant hCSP-L115 or L116 in

heterozygous $hsc4^{\Delta356}$ deletion mutants significantly suppressed their lethality ($p<0.0001$; *Figure 9F*). Hence, Hsc4 contributes to the toxicity of *CLN4* mutations.

Next, we examined to what degree altered Hsc4 levels may attenuate the oligomerization of *CLN4* mutant hCSPα. Hsc4 OE had no significant effects on the amount of hCSP-L115 and -L116 oligomers, monomers and overall protein levels ($p>0.05$; *Figure 9—figure supplement 1A–D*). However, reducing Hsc4 levels by expressing mutant hCSP-L115 or L116 in heterozygous $hsc4^{\Delta356}$ deletion mutants significantly reduced the amount of hCSP-L115 and -L116 oligomers ($p<0.02$; *Figure 9B–C,G*) but had no differential effects ($p>0.8$). It also reduced the levels of lipidated and non-lipidated hCSP-L115 and -L116 monomers, as well as overall protein levels ($p<0.05$; *Figure 9H–J*). The overall reduction of *CLN4* mutant hCSPα could have been due to a general dependence of hCSPα expression on normal Hsc4 levels. However, this is unlikely the case since reducing Hsc4 levels had no effect on the levels of monomeric WT hCSPα (*Figure 9K*), which was consistent with the normal dCSP levels observed in *hsc4* deletion mutants (*Bronk et al., 2001*).

Reducing Hsc4 levels also suppressed the amount of hCSP-L115 and -L116 protein accumulations on endosomes in the larval VNC ($p<0.05$; *Figure 9L* and *Figure 9—figure supplement 1E*). It also suppressed the amount of ubiquitinated protein accumulations induced by hCSP-L116 ($p<0.05$) but had no effect on L115-induced levels ($p=0.8$; *Figure 9M*). Reducing *hsc4* gene dosage had no effect on the synaptic levels of hCSP-L116 at larval NMJs ($p=0.7$; *Figure 9N*). However, it further lowered the synaptic levels of hCSP-L115 ($p<0.01$; *Figure 9N*). The underlying cause for these differential effects is unclear.

In conclusion, the suppression effects of a reduced *hsc4* gene dosage on the oligomerization and abnormal accumulation of *CLN4* mutant hCSPα on endosomes suggest that Hsc4 significantly contributes to this pathology. Since HA-tagged Hsc4 colocalized with the abnormal accumulations of hCSP-L115 and -L116 (*Figure 9O*), it may interfere with the prelysosomal trafficking and subsequent degradation of mutant hCSPα. Consistently, lower Hsc4 levels reduced the overall protein levels of mutant but not WT hCSPα (*Figure 9J–K*).

## Discussion

To better understand pathological mechanisms underlying *CLN4*, we developed two *Drosophila* models by expressing either *CLN4* mutant hCSPα or dCSP in neurons. Neuronal expression of either mutant hCSPα or dCSP mirrored key pathological features of post-mortem human *CLN4* brains, including reduced levels of palmitoylated hCSPα monomers and excessive formation of oligomers. Consistent with the idea that ubiquitination is likely a consequence of oligomerization, only ubiquitinated mutant hCSPα oligomers were detected but no ubiquitinated monomers. In general, the L115 mutation had stronger effects than L116, which has also been observed in post mortem brains and mammalian cell cultures (*Diez-Ardanuy et al., 2017*; *Greaves et al., 2012*; *Henderson et al., 2016*; *Zhang and Chandra, 2014*). Accordingly, expression of *CLN4* mutant hCSPα or dCSP in fly neurons provides a valid *CLN4* disease model.

### Dose-dependency of phenotypes in the *Drosophila CLN4* model

To assess the dose-dependency of *CLN4* phenotypes in the fly model, we controlled the gene dosage of transgenically expressed *CLN4* mutant hCSPα by neuronally expressing either one or two mutant transgenes in the presence of two endogenous WT dCSP gene copies.

Single copy expression of *CLN4* mutant hCSPα did not cause detectable degeneration or lethality, even though significant levels of ubiquitinated hCSPα oligomers and abnormal prelysosomal accumulations of hCSPα were present. However, doubling the gene dosage reduced the viability of *CLN4* mutant flies, which is likely due to neurodegeneration as indicated by fragmentated nuclear envelopes and 'bloated' Golgi cisternae of larval neurons (*Figure 6B*). The significant drop in viability indicated a sharp threshold of neurotoxicity, which was correlated with an increase in the levels of hCSPα oligomers and prelysosomal accumulations. Consistent with a neuronal failure, adult escapers showed reduced locomotor activity.

A similar dose-dependent effect was observed for the eye phenotype induced by GMR-driven expression of *CLN4* mutant hCSPα. The build-up of these pathological features in the *CLN4* fly model agrees with the assumed general concept of late onset neurodegenerative diseases, which typically show a progressive accumulation of protein aggregates and defects in protein homeostasis

over a long period that is met by a declining age-dependent capacity of protein homeostasis (*Labbadia and Morimoto, 2015*).

## *CLN4* mutations impair hCSPα's synaptic localization

Previous studies speculated that *CLN4* mutations may disrupt the trafficking and/or SV localization of hCSPα (*Benitez et al., 2011*; *Greaves et al., 2012*; *Nosková et al., 2011*), mainly because they reside in CSP's palmitoylated CS domain. The *CLN4* fly model confirms this prediction in two ways: First, *CLN4* mutations reduced the synaptic levels of mutant hCSPα. Second, they caused abnormal accumulations of mutant hCSPα in axons and neuronal somata. Analogous *CLN4* mutations in fly dCSP had similar effects excluding the possibility that these defects are the consequence of expressing a human protein in fly neurons.

In principle, there are two possibilities how *CLN4* mutations may mislocalize CSP: 1) *CLN4* mutations may primarily impair palmitoylation of the CS domain, which is required for ER and Golgi exit, and for a stable association with SVPs and/or SVs (*Greaves and Chamberlain, 2006*; *Greaves et al., 2008*; *Ohyama et al., 2007*; *Stowers and Isacoff, 2007*). However, this possibility is unlikely the case since mutant hCSPα was not detectable in the ER or Golgi of fly neurons. In addition, levels of non-lipidated *CLN4* mutant hCSPα were similar to WT hCSPα when expressed from one transgenic copy. 2) *CLN4* mutations may primarily induce oligomerization by exaggerating the dimerization properties of the CS domain (*Xu et al., 2010*) such that self-association leads mostly to misfolded high-molecular weight oligomers. Notably, oligomerization of *CLN4* mutant hCSPα can be suppressed by alanine or leucine substitutions of a cluster of palmitoylated cysteines (C122-125) located next to the L115/116 mutations (*Diez-Ardanuy et al., 2017*), which is consistent with the idea that *CLN4* mutations primarily induce oligomerization.

In fly neurons, the oligomerization of mutant hCSPα is apparently triggered after exiting the Golgi, either during its association with SVPs, or after arriving at axon terminals and associating with SVs. The transition of SVP cargo onto SVs or endosomes is not well understood but in principle SVPs can fuse with SVs, the synaptic plasma membrane, and endosomes (*Rizzoli, 2014*). In contrast, SVs are linked to endosomes either by ultrafast endocytosis, or the ability of freshly endocytosed individual SVs to fuse with endosomes (*Hoopmann et al., 2010*; *Soykan et al., 2017*; *Watanabe and Boucrot, 2017*; *Watanabe et al., 2013a*; *Watanabe et al., 2013b*). Mutant hCSPα oligomers are potentially re-routed from both SVPs and SVs because some mutant hCSPα colocalizes with SVs at axon terminals. Nevertheless, re-routing of ubiquitinated oligomers is likely the main factor reducing synaptic levels of *CLN4* mutant hCSPα.

The ubiquitination of *CLN4* mutant hCSPα oligomers likely indicates that they are misfolded and thus targets for ubiquitination and degradation like other misfolded proteins (*Wang et al., 2017*). Consistently, only ubiquitinated mutant hCSPα oligomers but no ubiquitinated monomers were detected in fly neurons. Since WT dCSP (*Figure 5—figure supplement 1*) and hCSPα are mostly degraded by lysosomes (*Benitez and Sands, 2017*; *Chapel et al., 2013*; *Sambri et al., 2017*; *Schröder et al., 2007*; *Tharkeshwar et al., 2017*), one expects that ubiquitinated mutant hCSPα oligomers are sorted into the endolysosomal pathway if they remain associated with membranes. This appears to be the case since the majority of abnormal mutant hCSPα accumulations in fly neurons represented prelysosomal endosomes, which were defined by the colocalization of ubiquitinated proteins and the prelysosomal markers LAMP1-GFP and HRS. Finally, the suppression of mutant hCSPα phenotypes by reduced dCSP or Hsc4 levels revealed a strong link between the oligomerization of mutant hCSPα, its re-routing onto prelysosomal membranes, and its induced lethality (*Figures 7–8*). Hence, these data suggest that *CLN4* mutations may primarily induce oligomerization of membrane-associated hCSPα, which causes ubiquitination and re-routing into lysosomal pathways.

## *CLN4* mutations impair prelysosomal trafficking of hCSPα

Axonal prelysosomal endosomes containing mutant hCSPα are likely retrogradely trafficked for lysosomal processing since degradative lysosomes containing cathepsin B/D are essentially absent from distal axons (*Cai et al., 2010*; *Cheng et al., 2018*; *Gowrishankar et al., 2015*). Both the axonal trafficking as well as the maturation of prelysosomal endosomes containing mutant hCSPα are potentially impaired. A physical defect in axonal trafficking is indicated by the large size of the hCSPα

accumulations and by the frequent occurrence of large and abnormal EM-dense membrane structures in axons of *CLN4* mutants, which have the potential to 'clog' axons.

A reduced efficiency in the maturation of mutant hCSPα-positive prelysosomal endosomes is indicated by their abnormally large size and by their failure to co-label with the PI$_3$P lipid reporter FYVE-GFP. The latter may be due to two alternatives: First, hCSPα-positive endosomes may have normal PI$_3$P levels but most of it is bound by PI$_3$P-binding proteins. At least in part, this could be facilitated by HRS (*Raiborg et al., 2001a*), which strongly co-labels with hCSPα accumulations. Alternatively, PI$_3$P could be depleted from these endosomal membranes, which then would imply that HRS is abnormally retained.

Since only a small number of mutant hCSPα accumulations co-localized with the early endosome marker Rab5 but essentially no accumulations colocalized with the late endosomal or lysosomal markers Rab7 and spinster, it is possible that mutant hCSPα may impair the transition from early to late endosomes.

The frequent occurrence of abnormal, multi-laminar structures containing EM-dense membranes and abnormal autophagosome-like structures further indicate that *CLN4* mutations impair steps of membrane trafficking in axons and somata. Since similar ultrastructural defects were observed after disruption of ESCRT function (*Doyotte et al., 2005*; *Lee et al., 2007*), the abnormal endosomal membranes of *CLN4* mutants suggest a defect in prelysosomal trafficking. This is further supported by the accumulation of mutant hCSPα on prelysosomal membranes including ATG8/LC3-GFP positive autophagosomes. Alternatively, at least some of the abnormal membrane structures of *CLN4* mutants could be the consequence of defects in the unconventional secretion of misfolded cytosolic proteins, which is facilitated by CSP proteins at least in non-neuronal cells (*Xu et al., 2018*).

The *CLN4* fly model also provides evidence for a general protein ubiquitination defect. Both of the *CLN4* mutations caused an increase in ubiquitinated protein accumulations in axons and neuronal somata that were not associated with accumulations of *CLN4* mutant hCSPα. Consistent with an altered protein homeostasis, significant proteomic and lipidomic changes were found in *CLN4* mutant post-mortem brains (*Henderson et al., 2016*). Even though *CLN4* mutant CSPα's ability to stimulate the ATPase activity of Hsc70 progressively deteriorates in vitro (*Zhang and Chandra, 2014*), the excessive ubiquitination phenotype of the fly model is unlikely due to a reduced CSP/Hsc70 chaperone function since reducing WT dCSP levels suppressed the ubiquitination effects. Hence, the excessive ubiquitination may be a secondary consequence of the prelysosomal trafficking defects induced by *CLN4* mutations.

## *CLN4* alleles genetically resemble hypermorphic gain of function mutations

Previously, it has been suggested that the progressive oligomerization of *CLN4* mutant hCSPα may cause a dominant-negative effect by sequestering and depleting WT CSPα (*Greaves et al., 2012*; *Henderson et al., 2016*; *Nosková et al., 2011*; *Zhang and Chandra, 2014*). However, *CLN4* mutant hCSPα had no effect on the synaptic and overall levels of endogenous WT dCSP in the fly model. In addition, decreasing or even abolishing WT dCSP reduced oligomer formation of mutant hCSPα. Both effects are inconsistent with a dominant-negative effect of L115 and L116 mutations. In addition, genetic analysis of other phenotypes found no evidence supporting dominant-negative effects in the *CLN4* fly model. Instead, this analysis suggests that most of the observed phenotypes are due to gain of function effects.

Reducing or abolishing WT dCSP levels suppressed the oligomerization and endosomal accumulation of *CLN4* mutant hCSPα, the elevated protein ubiquitination, and the premature lethality. Increasing WT dCSP or hCSPα levels enhanced the oligomerization and lethality of *CLN4* mutations. Taken together, these results suggest that the respective phenotypes are due to a hypermorphic gain of function mutation that increases an activity of hCSPα. A similar dependence of *CLN4* phenotypes on increased WT CSPα levels was observed in primary fibroblasts of CSPα-deficient mice after co-expressing WT CSPα with CSPα-L115R, which increased levels of CSPα oligomers and lysosomal lysotracker signals (*Benitez and Sands, 2017*).

Notably, there are also *CLN4* phenotypes of the fly model that could not be genetically classified in an unambiguous manner. These phenotypes include the effects of altered dCSP levels on monomeric and synaptic levels of *CLN4* mutant hCSPα, which were either not affected or further reduced by lower dCSP levels.

The primary effect of the gain of function mutation induced by L115 and L116 is not known. However, the fly model revealed a correlation among the degree of mutant hCSPα oligomerization, accumulation on endosomes, and lethality. Accordingly, it seems possible that *CLN4* mutations primarily drive the excessive oligomerization of hCSPα by exacerbating the known dimerization properties of the CS domain (*Xu et al., 2010*). Such an effect would explain the excessive routing of the likely misfolded oligomers into prelysosomal pathways and the depletion of mutant hCSPα at synapses. The impaired processing of the endosomes containing accumulating mutant hCSPα is then potentially a secondary effect of misfolded oligomers.

Another possibility for the gain of function effect arises from the recently described role of hCSPs on late endosomes for an unconventional secretion pathway, termed misfolding-associated protein secretion (MAPS) (*Xu et al., 2018*). Even though it is unclear whether neuronal hCSPα participates in MAPS, the gain of function effect could exacerbate the trafficking of hCSPα onto late endosomes for MAPS after exiting the Golgi. Together, an excessive routing onto endosomes and oligomerization of mutant hCSPα may then impair membrane trafficking pathways requiring late endosomes like the lysosomal and MAPS pathways.

The dependence of key *CLN4* phenotypes on WT CSP levels paralleled the modulatory effects of hCSPα's main interaction partner, Hsc70. Similar to reducing WT dCSP levels, reducing levels of Hsc4 by one gene copy suppressed the lethality and eye phenotypes induced by *CLN4* mutant hCSPα, its oligomerization and endosomal accumulation. These modulatory effects are driven either by an individual activity of Hsc70 itself, or by a synergistic activity of the CSP/Hsc70 complex. Neither of these activities are necessarily localized to SVs. In support of the latter, co-expressed HA-tagged Hsc70-4 co-localized with the accumulations of *CLN4* mutant hCSPα on endosomes, which may interfere with the dissociation of clathrin coats that are anchored by the ESCRT component HRS (*Raiborg et al., 2001b*). Alternatively, the respective Hsc70 activity may be associated with endosomal microautophagy (*Sahu et al., 2011*; *Uytterhoeven et al., 2015*) or the unconventional secretion of misfolded cytosolic proteins (*Xu et al., 2018*). At least for this fly model, an activity of Hsc70 associated with chaperone-mediated autophagy (*Kaushik and Cuervo, 2012*) can be excluded since the critical lysosomal translocator LAMP-2 is absent in flies (*Uytterhoeven et al., 2015*).

## Concluding remarks

This initial study of the *CLN4* fly model revealed a number of novel and important insights into the pathology of *CLN4*, most notably, the unexpected genetic nature of *CLN4* mutations, the mislocalization of mutant hCSPα, a general ubiquitination defect, and prelysosomal processing defects. The modulating effects of altered WT CSP or Hsc70 levels on key *CLN4* phenotypes provided a strong correlation between the premature lethality of *CLN4* mutant flies, oligomerization and accumulation of mutant hCSPα on prelysosomal endosomes. Accordingly, the neurotoxicity of *CLN4* mutant hCSPα may be at least in part due to the formation of ubiquitinated hCSPα oligomers that then progressively accumulate on prelysosomal endosomes and interfere with their processing for lysosomal fusion and potentially their retrograde axonal trafficking.

The pathological mechanisms underlying *CLN4* appear to be different from those of most other NCLs, which are typically due to loss of function mutations in genes that mediate lysosomal function, ER-lysosomal trafficking, or protein lipidation (*Cárcel-Trullols et al., 2015*; *Cooper et al., 2015*; *Cotman et al., 2013*; *Mole and Cotman, 2015*; *Warrier et al., 2013*). As such, the pathological mechanisms underlying *CLN4* appear similar to those of 'classical' neurodegenerative diseases that are induced by a progressive build-up of protein oligomers/aggregates that then leads to a failure of protein homeostasis and/or lysosomal pathways (*Abramov et al., 2009*; *Labbadia and Morimoto, 2015*; *Lansbury and Lashuel, 2006*; *Muchowski and Wacker, 2005*; *Neefjes and van der Kant, 2014*). While we are just beginning to understand the complicated nature of *CLN4*, the fly model provides a valuable tool for future work to dissect pathological mechanisms underlying *CLN4*.

# Materials and methods

## Key resources table

| Reagent type (species) or resource | Designation | Source or reference | Identifiers | Additional information |
|---|---|---|---|---|
| Gene (*Drosophila melanogaster*) | csp | Flybase | FLYB:FBgn0004179 | Cysteine string protein |
| Gene (*human*) | DNAJC5 | NCBI | GeneID:80331 | Encodes CSPα |
| Transcript (*Drosophila melanogaster*) | dCSP-2/CSP-PC | NCBI Genbank | NM_168950.4 | Reference Sequence used for synthesis of dCSP constructs, ORF: 186–872 |
| Recombinant DNA reagent | pBID-UASC | *Wang et al., 2012* | RRID:Addgene_35200 | PhiC31 attB 10xUAS vector for site specific recombinase insertions |
| Recombinant DNA reagent | pUC57-dCSP2.WT | Genscript; This paper | | Full length wildtype dCSP2 ORF with Kozak sequence and NotI/KpnI restrictions sites cloned into pUC57 |
| Recombinant DNA reagent | pUC57-dCSP2.V117R | Genscript; This paper | | As above with V117R mutation |
| Recombinant DNA reagent | pUC57-dCSP2.I118Δ | Genscript; This paper | | As above with I118Δ mutation |
| Recombinant DNA reagent | PGEX-KG_rCSPα | *Zhang and Chandra, 2014* | | Wildtype rat CSPα cDNA sequence with V112F point mutation to match human aa sequence |
| Recombinant DNA reagent | PGEX-KG_ hCSPα.L115R | *Zhang and Chandra, 2014* | | As above with L115R mutation |
| Recombinant DNA reagent | PGEX-KG_ hCSPα.L116Δ | *Zhang and Chandra, 2014* | | As above with L116Δ mutation |
| Recombinant DNA reagent | pBID-UASC_ CSPα.WT | This study | | full length wildtype ORF encoding rat/human CSPα cloned into pBID-UASC transformation vector |
| Recombinant DNA reagent | pBID-UASC_ CSPα.L115R | This study | | As above with L115R mutation |
| Recombinant DNA reagent | pBID-UASC_ CSPα.L116Δ | This study | | As above with L116Δ mutation |
| Recombinant DNA reagent | pBID-UASC_ dCSP2.WT | This study | | Full length wildtype ORF encoding dCSP2 cloned into pBID-UASC transformation vector |
| Recombinant DNA reagent | pBID-UASC_ dCSP2.V117R | This study | | As above with V117R mutation |
| Recombinant DNA reagent | pBID-UASC_ dCSP2.I118Δ | This study | | As above with I118Δ mutation |
| Sequenced-based reagent | rCSP NotI for | Sigma | | 5'-GAGCGGCC GCCAAAATG GCTGACCAGAGG CAGCGCTC-3' |

*Continued on next page*

*Continued*

| Reagent type (species) or resource | Designation | Source or reference | Identifiers | Additional information |
|---|---|---|---|---|
| Sequenced-based reagent | rCSP KpnI rev | Sigma | | 5'-CATGGTACCT TAGTTGA ACCCGTCGGTGT GATAGCTGG-3' |
| Genetic reagent (*D. melanogaster*) | w[1118] | Caltech collection: Seymour Benzer, Caltech | FlyB:FBal0018186 | Genotype: w[1118]; isogenized genetic background |
| Genetic reagent (*D. melanogaster*) | elav::Gal4[C155] | Bloomington Drosophila Stock Center | FlyB:FBst0000458; RRID:BDSC_458 | Genotype: P{w[+mW.hs]=GawB} elav[C155] |
| Genetic reagent (*D. melanogaster*) | nSyb::Gal4 | Hugo Bellen, Baylor College of Medicine | FlyB:FBst0051635; RRID:BDSC_51635 | Genotype: y[1] w[*]; P{w[+m*]=nSyb-GAL4.S}3 |
| Genetic reagent (*D. melanogaster*) | GMR::Gal4 | Bloomington Drosophila Stock Center | FlyB:FBst0001104; RRID:BDSC_1104 | Genotype: w[*]; P{w[+mC]=GAL4 ninaE. GMR}12 |
| Genetic reagent (*D. melanogaster*) | UAS::nsyb-EGFP | Bloomington Drosophila Stock Center | FlyB:FBst0006921; RRID:BDSC_6921 | Genotype: w[*]; P{w[+mC]=UAS nSyb. eGFP}2 |
| Genetic reagent (*D. melanogaster*) | UAS::GFP-myc-2xFYVE | Bloomington Drosophila Stock Center | FlyB:FBst0042712; RRID:BDSC_42712 | Genotype: w[*]; P{w[+mC]=UAS-GFP-myc-2xFYVE}2 |
| Genetic reagent (*D. melanogaster*) | UAS::TSG101 RNAi | Bloomington Drosophila Stock Center | FlyB:FBst0035710; RRID:BDSC_35710 | Genotype: y[1] sc[*] v[1] sev[21]; P{y[+t7.7] v[+t1.8]=TRiP. GLV21075}attP2 |
| Genetic reagent (*D. melanogaster*) | UAS:GFP-KDEL | Bloomington Drosophila Stock Center | FlyB:FBst0009898; RRID:BDSC_9898 | Genotype: w[*]; P{w[+mC]=UAS GFP. KDEL}11.1 |
| Genetic reagent (*D. melanogaster*) | UAS::hLAMP1-GFP | Bloomington Drosophila Stock Center | FlyB:FBti0150347; RRID:BDSC_42714 | Genotype: w[*]; P{w[+mC]=UAS-GFP-LAMP}2; P{w[+m*]=nSyb-GAL4.S}3/ T(2;3)TSTL, CyO: TM6B, Tb[1] |
| Genetic reagent (*D. melanogaster*) | UAS::GFP-Rab5 | Bloomington Drosophila Stock Center | FlyB:FBst0043336; RRID:BDSC_43336 | Genotype: w[*]; P{w[+mC]=UAS-GFP-Rab5}3 |
| Genetic reagent (*D. melanogaster*) | UAS::spin-myc.EGFP | Bloomington Drosophila Stock Center | FlyB:FBst0039668; RRID:BDSC_39668 | Genotype: w[*]; P{w[+mC]=UAS spin. myc-EGFP}B |
| Genetic reagent (*D. melanogaster*) | UAS::GFP-LC3(ATG8) | Bloomington Drosophila Stock Center | FlyB:FBst0008730; RRID:BDSC_8730 | Genotype: w[*]; P{w[+mC]=UASp-eGFP-huLC3}1; P{w[+mC]=GAL4:: VP16-nos. UTR}CG6325[MVD1] |
| Genetic reagent (*D. melanogaster*) | UAS::syx-1A | Bloomington Drosophila Stock Center | FlyB:FBst0051619; RRID:BDSC_51619 | Genotype: y[1] w[*]; P{w[+mC]=UAS-Syx1A.B}6 |
| Genetic reagent (*D. melanogaster*) | UAS::hsc70-4 KD1 | Bloomington Drosophila Stock Center | FlyB:FBst0054810; RRID:BDSC_54810 | Genotype: y[1] v[1]; P{y[+t7.7] v[+t1.8]=TRiP. HMJ21529}attP40 |

*Continued on next page*

Continued

| Reagent type (species) or resource | Designation | Source or reference | Identifiers | Additional information |
|---|---|---|---|---|
| Genetic reagent (*D. melanogaster*) | UAS::hsc70-4 KD2 | Bloomington Drosophila Stock Center | FlyB:FBst0028709; RRID:BDSC_28709 | Genotype: y[1] v[1]; P{y[+t7.7] v[+t1.8]=TRiP. JF03136}attP2 |
| Genetic reagent (*D. melanogaster*) | UAS::hsc70-4 KD3 | Bloomington Drosophila Stock Center | FlyB:FBst0034836; RRID:BDSC_34836 | Genotype: y[1] sc[*] v[1]; P{y[+t7.7] v[+t1.8]=TRiP. HMS00152} attP2/TM3, Sb[1] |
| Genetic reagent (*D. melanogaster*) | attP-22A | Bloomington Drosophila Stock Center | FlyB:FBst0024481; RRID:BDSC_24481 | Genotype: y[1] M {vas-int.Dm} ZH-2A w[*]; M {3xP3-RFP.attP'} ZH-22A PhiC31 Insertion background, all generated lines outcrossed into w[1118] |
| Genetic reagent (*D. melanogaster*) | UAS::venus-Rab7.WT | R Hisinger (Freie Universität, Berlin, Germany); *Cherry et al., 2013* | FlyB:FBal0294208 | |
| Genetic reagent (*D. melanogaster*) | hsc70-4[Δ356] | *Bronk et al., 2001* | FlyB:FBal0124174 | Genotype: w[1118];; hsc70-4[Δ356] |
| Genetic reagent (*D. melanogaster*) | UAS::Hsc70-4 | Karen Palter, Temple University; *Elefant and Palter, 1999,* | FlyB:FBst0005846; RRID:BDSC_5846 | Genotype: w[126]; P{w[+mC]=UAS-Hsc70-4.WT}B |
| Genetic reagent (*D. melanogaster*) | UAS:HA-Hsc70-4.WT | P Verstreken; *Uytterhoeven et al., 2015* | FlyB:FBal0318413 | Genotype: w[1118]; P{w[+mC]=UAS-Hsc70-4.HA} |
| Genetic reagent (*D. melanogaster*) | UAS::SNAP25 | Bloomington Drosophila Stock Center | FlyB:FBst0051997; RRID:BDSC_51997 | Genotype: y[1] w[*]; P{w[+mC]=UAS-Snap25.L}9 |
| Genetic reagent (*D. melanogaster*) | UAS::SNAP25 KD | Bloomington Drosophila Stock Center | FlyB:FBst0027306; RRID:BDSC_27306 | Genotype: y[1] v[1]; P{y[+t7.7] v[+t1.8]=TRiP. JF02615}attP2 |
| Genetic reagent (*D. melanogaster*) | csp[R1] | *Zinsmaier et al., 1994* | FlyB:FBst0032035; RRID:BDSC_32035 | Genotype: w[1118]; csp[X1]/TM6Tb[1]Sb[1]; partial deletion of csp (genetic null) |
| Genetic reagent (*D. melanogaster*) | csp[X1] | *Zinsmaier et al., 1994* | FlyB:FBst0051998; RRID:BDSC_51998; | Genotype: w[1118]; csp[R1]/TM6Tb[1]Sb[1]; complete deletion of csp locus, maintained in this lab |
| Genetic reagent (*D. melanogaster*) | UAS::dCSP2 | this study | | Genotype: w[1118]; M{UAS-dCSP2}ZH-22A |
| Genetic reagent (*D. melanogaster*) | UAS::dCSP2.V117R | this study | | Genotype: w[1118]; M{UAS-dCSP2.V117R} ZH-22A |
| Genetic reagent (*D. melanogaster*) | UAS::dCSP2.I118Δ | this study | | Genotype: w[1118]; M{UAS-dCSP2.I118Δ} ZH-22A |
| Genetic reagent (*D. melanogaster*) | UAS::hCSPα | this study | | Genotype: w[1118]; M{UAS-hCSPα}ZH-22A |
| Genetic reagent (*D. melanogaster*) | UAS::hCSPα.L115R | this study | | Genotype: w[1118]; M{UAS-hCSPα.L115R} ZH-22A |

*Continued on next page*

*Continued*

| Reagent type (species) or resource | Designation | Source or reference | Identifiers | Additional information |
|---|---|---|---|---|
| Genetic reagent (*D. melanogaster*) | UAS::hCSPα.L116Δ | this study | | v: w[1118]; M{UAS-hCSPα. L116R}ZH-22A |
| Antibody | anti-CSPα (rabbit polyclonal) | Enzo Life Sciences | Cat#: VAP-SV003E; RRID:AB_2095057 | Immunostaining (IS, 1:2000); Western Blot (WB, 1:20000); IP (1:250) |
| Antibody | anti-dCSP (mouse monoclonal) | *Zinsmaier et al., 1990* | Cat# DSHB:DCSP-1 (ab49); RRID:AB_2307345 | IS (1:20); WB (1:250) |
| Antibody | anti-GFP (mouse monoclonal) | Developmental Studies Hybridoma Bank | Cat# DSHB: GFP-12A6; RRID:AB_2617417 | IS (1:1000) |
| Antibody | anti-HRS (guinea-pig polyclonal) | HJ Bellen, Baylor College of Medicine, Houston, TX; *Lloyd et al., 2002* | | IS (1:2000); WB (1:2000) |
| Antibody | anti-HA (rat monoclonal) | Roche | Cat#: 3F10; RRID:AB_2314622 | IS (1:200) |
| Antibody | anti-GM130 (rabbit polyconal) | Abcam | Cat# ab31561, RRID:AB_2115328 | IS (1:200) |
| Antibody | anti-Golgin245 (goat polyclonal) | Developmental Studies Hybridoma Bank | Cat#: Golgin245; RRID:AB_2618260 | IS (1:2000) |
| Antibody | anti-GMAP (goat polyclonal) | Developmental Studies Hybridoma Bank | Cat#: GMAP; RRID:AB_2618259 | IS (1:2000) |
| Antibody | anti-Rab7 (mouse monoclonal) | Developmental Studies Hybridoma Bank | Cat#: Rab7; RRID:AB_2722471 | IS (1:100) |
| Antibody | anti-Syx1A (mouse monoclonal) | Developmental Studies Hybridoma Bank | Cat#: 8C3; RRID:AB_528484 | IS (1:200) |
| Antibody | anti-β-tubulin (mouse monoclonal) | Developmental Studies Hybridoma Bank | Cat#: E7; RRID:AB_528499 | WB (1:1000) |
| Antibody | anti-conjugated-ubiquitin (mouse monoclonal) | Enzo Life Sciences | Cat#: BML-PW8810; RRID:AB_10541840; Clone FK2 | IS (1:2000); WB (1:2000) |
| Antibody | anti-HRP Alexa Fluor 647-conjugated (goat polyclonal) | Jackson Immuno Research Labs | Cat#: 123-605-021; RRID:AB_2338967 | IS (1:500) |
| Antibody | anti-mouse IgG1 AlexaFluor 488-conjugated (goat polyclonal) | Thermo Fisher Scientific | Cat#: A-21121; RRID:AB_2535764 | IS (1:500) |
| Antibody | anti-rabbit IgG (H+L) Cy3-conjugated (donkey polyclonal) | Jackson Immuno Research Labs | Cat#: 711-165-152; RRID:AB_2307443 | IS (1:500) |
| Antibody | anti-guinea pig IgG (H+L) Alexa Fluor 488-conjugated (goat polyclonal) | Thermo Fisher Scientific | Cat#: A-1107; RRID:AB_2534117 | IS (1:1000) |

*Continued on next page*

*Continued*

| Reagent type (species) or resource | Designation | Source or reference | Identifiers | Additional information |
|---|---|---|---|---|
| Antibody | anti-rat IgG (H+L) Alexa Fluor 488-conjugated (goat polyclonal) | Jackson Immuno Research Labs | Cat#: 112-545-167; RRID:AB_2338362 | IS (1:500) |
| Antibody | anti-rabbit IgG (H+L) HRP-conjugated (goat polyclonal) | Thermo Fisher Scientific | Cat#: A16096; RRID:AB_2534770 | WB (1:10000) |
| Antibody | anti-mouse IgG HRP-conjugated (goat polyclonal) | Thermo Fisher Scientific | Cat#: 32430; RRID:AB_1185566 | WB (1:5000) |
| Commercial assay or kit | Western Clarity ECL kit | BioRad | Cat#: 1705061 | |
| Commercial assay or kit | A/G PLUS-Agarose Beads | Santa Cruz Biotechnology | Cat#: sc-2003 | |
| Software, algorithm | FIJI/ImageJ | NIH | RRID:SCR_002285 | |
| Software, algorithm | Prism | Graphpad | RRID:SCR_002798 | |
| Software, algorithm | Adobe Photoshop CC | Adobe | RRID:SCR_014199 | |
| Software, algorithm | Quantity One 1-D Analysis Software | BioRad | RRID:SCR_014280 | |

## *Drosophila* strains and husbandry

All flies were raised at 23°C on standard cornmeal culture media with a 12/12 light-dark cycle unless otherwise specified. Gal4 driver strains nSyb-Gal4, elav-Gal4(C155) and GMR-Gal4, and UAS strains expressing EGFP-nSyb, GFP-myc-2xFYVE, TSG101 KD, hLAMP1-GFP, Spin-myc-EGFP, GFP-LC3 (ATG8), GFP-KDEL, nSyb-EGFP, Rab5-YFP, Rab7-YFP, Hsc70-4 KD (1-3), Syx1A, SNAP25, and SNAP25 KD were obtained from the Bloomington *Drosophila* Stock Center (BDSC, Bloomington, Indiana). UAS strains expressing Venus-Rab7 (*Cherry et al., 2013*), Hsc70-4 (*Elefant and Palter, 1999*), and HA-Hsc70-4 (*Uytterhoeven et al., 2015*) were obtained from P.R. Hiesinger (Freie Universität, Berlin, Germany), K. Palter (Temple University, Philadelphia, PA) or P. Verstreken (VIB Center for the Biology of Disease, Leuven, Belgium), respectively. The *csp* (R1, X1) and *hsc70-4* (Δ356) gene deletion alleles were generated previously by us (*Bronk et al., 2001*; *Zinsmaier et al., 1994*).

## Generation of UAS transgenes

To generate transgenes that individually express human WT, L115R- and L116Δ-mutant CSPα under the transcriptional control of the Gal4/UAS system (*Brand and Perrimon, 1993*), the open reading frames of modified rat cDNAs encoding human WT and mutant CSPα (*Zhang and Chandra, 2014*) were PCR amplified using a forward primer containing a NotI restriction site and a *Drosophila* consensus Kozak sequence (5' GAGCGGCCGCCAAAATGGCTGACCAGAGGCAGCGCTC 3') and a reverse primer containing a KpnI site just after the stop codon (5' CATGGTACCTTAGTTGAACCCG TCGGTGT GATAGCTGG 3'). The obtained PCR products were cleaved with NotI and KpnI and directionally cloned into a NotI/KpnI cleaved pBID-UASC vector (*Wang et al., 2012*). cDNAs encoding WT and *CLN4* mutant dCSP were synthesized de novo (GenScript, Piscataway, NJ) using cDNA sequences encoding the open reading frame of CSP2 (CSP-PC, accessionID: NM_168950.4)) as template (*Nie et al., 1999*). The mutations V117R and I118Δ were introduced into dCSP2 to generate *CLN4* mutant dCSP. Both V117R and I118Δ are analogous to the human *CLN4* mutations L115R and L116Δ, respectively. The synthesized cDNAs contained a NotI site followed by a Kozak sequence at the 5' end and a Kpn1 site after the stop codon at the 3' end and were inserted into pUC57 plasmids. For transgene expression, the cDNAs were then directionally subcloned into a pBID-UASC vector using the NotI and KpnI cleavage sites.

Transgenic animals were generated by φC31-based integration (*Bischof et al., 2007*) of the UAS transgenes into the attP site at 22A2 on chromosome 2 (2L:1476459..1476459). pBID-UASC-CSP-x plasmids were injected into *y¹ M[vas-int.Dm]ZH-22A w\*; M[3xP3-RFP.attP']ZH-22A* (BDSC #24481) embryos (Rainbow Transgenic Flies, Camarillo, CA). At least two independent recombinant strains were obtained for each transgene and out-crossed to exchange non-recombinant chromosomes. The *3xP3-RFP* cassette was removed by loxP-mediated recombination as described (*Bischof et al., 2007*). Homozygous strains containing UAS transgenes were established in a genetic background representing WT control (*w¹¹¹⁸*) and *dcsp* deletion mutants (*w¹¹¹⁸; dcsp^{X1}*).

Crosses for single copy transgene expression were made by crossing homozygous male *w¹¹¹⁸, elav-Gal4[C155]* flies to homozygous *w¹¹¹⁸; M[UAS::CSP-x.]ZH-22A* female flies yielding female F1 progeny expressing CSP-x (*w¹¹¹⁸, elav-Gal4[C155]/w¹¹¹⁸; M[UAS::CSP-x]ZH-22A /+*) and male progeny containing a silent (non-expressed) transgene (*w¹¹¹⁸; M[UAS::CSP-x]ZH-22A/+*). Crosses for two copy expression were made by crossing females *w¹¹¹⁸, elav-Gal4[C155]; M[UAS::CSP-x]ZH-22A/ CyO, Actin-GFP* to male *w¹¹¹⁸; M[UAS::CSP-x]ZH-22A*. Female F1 progeny heterozygous for the *elav* driver and homozygous for the UAS transgene (*w¹¹¹⁸, elav-Gal4[C155]/w¹¹¹⁸; M[UAS::CSP-x}ZH-22A*) were selected for analysis.

## dCSP null rescue

All genetic rescue experiments employing a *csp^{-/-}* deletion null genetic background expressed WT or *CLN4* mutant hCSPα from a single transgene. Female *w¹¹¹⁸, elav-Gal4[C155]; csp^{R1}/TM6 Tb Sb* flies were crossed to *w¹¹¹⁸; M[UAS::hCSP-x]ZH-22A; csp^{X1}/TM6 Tb Sb* males or *w¹¹¹⁸; csp^{X1}/TM6 Tb Sb* males for control. Flies were raised at 23°C. Male progeny (*w¹¹¹⁸, elav-Gal4[C155]; M[UAS::hCSP] ZH-22A/+; csp^{X1/R1}*) were used because of higher expression and better rescue. 10–15 freshly enclosed flies of the respective genotypes were cultured in separate vials and transferred weekly to fresh food. Dead flies were scored daily.

## Viability and adult lifespan

The following crossing scheme was used: For single copy expression, hemizygous *w¹¹¹⁸, elav-Gal4 [C155]* males were crossed to homozygous *w¹¹¹⁸; UAS::CSP-x* females yielding non-expressing F1 control males (*w¹¹¹⁸; UAS::CSP-x/+*) and CSP-expressing F1 females (*w¹¹¹⁸, elav-Gal4[C155]/w¹¹¹⁸; UAS::CSP-x/+*). For two copy expression, hemizygous *w¹¹¹⁸, elav-Gal4[C155]; UAS::CSP-x/CyO, Actin-GFP* males were crossed to homozygous *w¹¹¹⁸; UAS::CSP-x* females yielding non-expressing F1 males (*w¹¹¹⁸; UAS::CSP-x*) and CSP-expressing F1 females (*w¹¹¹⁸, elav-Gal4[C155]/w¹¹¹⁸; UAS:: CSP*). Flies were raised at 23°C or 28°C. Viability was determined by scoring the ratio of freshly enclosed females and male flies. Adult lifespan was determined by culturing freshly enclosed males and females in separate vials and scoring dead flies daily or every 2nd day. Adult flies were transferred weekly to fresh food.

## Immunostainings

Wandering 3rd instar larvae were dissected in Sylgard-coated dishes containing cold Ca²⁺ free HL3 solution (in mM: 70 NaCl, 5 KCl, 20 MgCl₂, 10 NaHCO₃, 5 Trehalose, 115 sucrose, 5 HEPES, pH 7.2). Dissected larvae were fixed for 20 (VNC) or 45 min (NMJs) in 4% formaldehyde solution (Electron Microscopy Sciences, Hatfield, PA) in phosphate-buffered saline (PBS), pH 7.3 at room temperature (RT). PBS (pH 7.3) supplemented with 0.2% Triton X-100 (PBST) was used for immunostainings of larval NMJs while PBS (pH 7.3) supplemented with 0.4% Triton X-100 was used for immunostainings of larval VNCs. After washing 3 times for 10 min in PBST at RT, larvae were incubated with primary antibodies diluted in PBST overnight at 4°C, washed 3x for 15 min at RT, incubated with secondary antibodies diluted in PBST for 2 hr at RT or overnight at 4°C and finally washed 3x with PBST for 15 min at RT. Confocal images were acquired the same day, otherwise preparations were post-fixed.

The following antibodies and dilutions were used: rabbit anti-CSPα, 1:2000 (Enzo Life Sciences Cat# VAP-SV003E, RRID:AB_2095057); mouse anti-dCSP, 1:250 (DSHB Cat# DCSP-1 (ab49), RRID: AB_2307345; *Zinsmaier et al., 1990*); mouse anti-GFP, 1:1000 (DSHB Cat# DSHB-GFP-12A6, RRID: AB_2617417); guinea pig anti-HRS, 1:2000; (*Lloyd et al., 2002*), H. Bellen, Baylor College of Medicine, Houston, TX); rat anti-HA, 1:200 (Roche Cat# 3F10, RRID:AB_2314622); rabbit anti-GM130, 1:200 (Abcam Cat# ab31561, RRID:AB_2115328); goat anti-Golgin245, 1:2000 (DSHB Cat#

Golgin245, RRID:AB_2618260), goat anti-GMAP, 1:2000 (DSHB Cat# GMAP, RRID:AB_2618259); mouse anti Rab7, 1:100 (DSHB Cat# Rab7, RRID:AB_2722471; *Riedel et al., 2016*); mouse anti-Syx1A, 1:200 (DSHB Cat# 8c3, RRID:AB_528484); mouse anti-ubiquitin-conjugated protein, 1:2000 (Enzo Life Sciences Cat# BML-PW8810, RRID:AB_10541840); goat anti-HRP Alexa Fluor 647-conjugated, 1:500 (Jackson ImmunoResearch Labs Cat# 123-605-021, RRID:AB_2338967); goat anti-mouse IgG1 Alexa Fluor 488-conjugated, 1:500 (Thermo Fisher Scientific Cat# A-21121, RRID:AB_2535764); donkey anti-rabbit IgG (H+L) Cy3-conjugated, 1:500 (Jackson ImmunoResearch Labs Cat# 711-165-152, RRID:AB_2307443); goat anti-guinea pig IgG (H+L) Alexa Fluor 488-conjugated, 1:1000 (Thermo Fisher Scientific Cat# A-11073, RRID:AB_2534117); goat anti-rat IgG (H+L) Alexa Fluor 488-conjugated, 1:500 (Jackson ImmunoResearch Labs Cat# 112-545-167, RRID:AB_2338362).

## Confocal imaging

Stained preparations were imaged with an Olympus microscope BX50WI equipped with a confocal laser scanner (FluoView 300), a 60X water-immersion objective (LUMPLFL; N.A., 0.9), a multi-argon (630), a green HeNe (430), and a red HeNe (630) laser. Optical sections in the vertical axis were acquired using 0.8 µm (NMJs) or 1.5 µm (VNC) intervals for optical sectioning using Fluoview software. Images were analyzed offline using ImageJ software (FIJI v1.50, NIH).

For quantification of fluorescence signals, control and mutant larvae were dissected in the same dish such that fixation and antibody incubation were performed identically. All samples were imaged with the same laser settings. Fluorescence intensity per area was determined from a region of interest (ROI) encompassing single synaptic boutons by using Image J Software.

For quantification of CSP accumulations in VNC, z-projections with maximal intensity of optical sections were generated and cropped to a defined volume of $60 \times 80 \times 45$ µm ($216,000$ µm$^3$). These image stacks represented the dorsal-most part of hemi-segments A4 and A5. After thresholding for background subtraction, ROIs were drawn around all visible CSP-positive punctae; for those appearing in multiple optical sections only the section with the brightest signal was used. ROI areas were compiled to compare the cumulative area among genotypes.

## Western blot analysis

Larval brains were dissected from wandering 3$^{rd}$ instar larvae in in phosphate-buffered saline (PBS, pH 7.3) and five brains transferred to 60 µL buffer (2% SDS, 10% Glycerol, 60 mM Tris pH 6.8, 0.005% Bromophenol Blue, and 100 mM DTT). Brains were homogenized with a p200 pipette tip, boiled for 5 min and centrifuged for 3 min at 2000 g. The soluble fraction was transferred to a fresh tube, boiled for 1 min and an equivalent of ~1.5 brains was immediately loaded onto a 10% acrylamide gel for SDS-PAGE at 80V (Mini-Protean Cell, BioRad, Hercules, CA). Separated proteins were blotted onto nitrocellulose membranes at 20 V for 6 min using an iBlot system (Invitrogen, Carlsbad, CA). After transfer, the blot was blocked for 30 min using 1% bovine serum albumin fraction V (#BP1600-100, Fisher Scientific) in PBS supplemented with 0.2% Tween-20, pH 7.3 (PBSTw). Blots were incubated with primary antibodies overnight at 4˚C, washed, and incubated with HRP-conjugated secondary antibodies for 2 hr at 4˚C. To normalize for protein loading, blots were stripped (#21059; ThermoFisher Scientific) for 15 min at RT and immunostained for β-tubulin. Blots were imaged using a BioRad Western Clarity ECL kit and ChemiDoc XRS imaging system. Protein band intensities were quantified via densitometry analysis with Quantity One software (Bio-Rad). Raw intensity values were normalized to the measurement of a β-tubulin loading control and expressed as the n-fold change to an appropriate control genotype. Antibodies were diluted in PBSTw and used at the following concentrations: rabbit anti-CSPα 1:20,000 (Enzo Life Sciences Cat# VAP-SV003E, RRID:AB_2095057); mouse anti-dCSP, 1:20 (DSHB Cat# DCSP-1 (ab49), RRID:AB_2307345; *Zinsmaier et al., 1990*); mouse anti-ubiquitin-conjugated protein, 1:2000 (Enzo Life Sciences Cat# BML-PW8810, RRID:AB_10541840); mouse anti-β-tubulin, 1:1000 (DSHB Cat# E7, RRID:AB_528499); goat anti-mouse IgG HRP-conjugated, 1:5000 (Thermo Fisher Scientific Cat# 32430, RRID:AB_1185566; goat anti-rabbit IgG (H+L) HRP-conjugated 1:10,000 (Thermo Fisher Scientific Cat# A16096, RRID:AB_2534770).

## Hydroxylamine treatment

Adult heads were homogenized in 2% SDS, 100 mM DTT, 60 mM Tris pH 6.8, and then treated with 0.5M hydroxylamine pH 7.0 or 0.5 M Tris pH 7.0 (final concentration) for 24 hr at RT before being diluted 3-fold using 1x Laemmli buffer. Samples were boiled for 5 min before use for SDS-PAGE.

## Immunoprecipitation

Young adult flies (1–3 day post-eclosion) were flash frozen in liquid $N_2$ and stored at −80°C. Heads were enriched by brushing frozen heads on double metal sieves under liquid $N_2$. Approximately 100 μL of heads were homogenized in 500 μL IP buffer containing: 1% Triton X-100, 30 mM Tris-HCl (pH 7.2), 150 mM KCl, 0.5 mM $MgCl_2$. Homogenates were repeatedly centrifuged at 22,000 rpm to remove insoluble debris. The cleared supernatant was incubated with rabbit anti-CSPα at 1:200 (f.c.) for 1 hr on a rotator at 4°C and then incubated with 50 μL protein A/G PLUS-Agarose (#sc-2003, Santa Cruz Biotechnology, Dallas, Texas) for 2 hr at 4°C. Agarose beads were spun down with a tabletop centrifuge at 2000 g and the supernatant was removed and stored. The beads were washed 4x with IP buffer by repeated resuspension and centrifugation. Beads were finally resuspended in Laemmli buffer, boiled for 5 min, centrifuged for 2 min and transferred to a fresh tube for SDS-PAGE.

## Electron microscopy

Larvae were rapidly dissected in 4% PFA and then fixed overnight in fresh fixative containing: 3% glutaraldehyde, 1.5% formaldehyde, 2 mM $CaCl_2$, 0.1M sodium cacodylate, pH 7.3. Samples were rinsed in 0.1M cacodylate before being fixed with 2% OsO4 for 2 hr. Next, samples were washed with HPLC grade $H_2O$ before being dehydrated with ethanol in a stepped series (30%, 50%, 70%, 90%, 95%,100%) followed by 100% acetone. Tissue was infiltrated with Durcupan ACM plastic embedding media (#14040, Electron Microscopy Sciences, Hatfield, PA) through progressive mixtures with acetone (25%, 50%, 75%, 100%), hardened at 60°C, trimmed and sectioned with a diamond knife (70 nm). Sections were poststained with 4% uranyl acetate and lead citrate. Images were obtained on a JEOL 1200EX with an AMT XR80M-B camera running AMT software. For publication, figures were compiled and prepared with Photoshop CC (Adobe). Contrast and intensity of images was minimally adjusted. Images were cropped as needed.

## Quantification of eye phenotypes

This analysis was essentially done as described by others with some modifications (*Papanikolopoulou and Skoulakis, 2011*; *Santa-Maria et al., 2015*). Briefly, eye phenotypes of 1–2 day old adults were digitally imaged using a stereomicroscope. A semi-quantitative phenotype assessment was achieved by serially coding the obtained images and blind scoring by several naive researchers. Each individual eye was given a relative roughness score in comparison with known scoring classes: (1) normal WT-like eye. (2) Slightly 'rough' eye surface, slight dis-colorization, and/or least reduction in eye size. (3) Rough eye surface, significant dis-colorization, disorganized ommatidia, and/or reduced in size. (4) Rougher eye surface; loss of organized ommatidia, malformed and fused; more pronounced reduction in eye size. (5) Roughest eye surface, most pronounced loss of organization, and/or most pronounced reduction in eye size. Scores were collected for at least eight individual flies per genotype.

## Data and statistical analysis

Data from at least three independent animals or experimental trials were used for statistical analysis. Data are represented as mean, and error bars represent SEM. Gaussian distribution of data was assessed using a D'Agostino and Pearson omnibus or Shapiro-Wilk normality test using Prism software (GraphPad Software). Statistical significance was assessed by either a paired or unpaired *t* test, Mann-Whitney, or one-way ANOVA (Kruskal-Wallis for non-parametric data) test with appropriate post-hoc tests using Prism software). P values < 0.05,<0.01, and <0.001 are indicated in graphs with one, two, and three asterisks, respectively.

## Acknowledgements

We thank Drs. Hugo J Bellen (Baylor College of Medicine, Houston TX, USA), P Robin Heisinger (Freie Universität, Berlin, Germany), Patrik Verstreken (VIB-KU Leuven Center for Brain and Disease Research, Leuven, Belgium), and the Developmental Studies Hybridoma Bank at the University of Iowa for antibodies and/or fly strains. We thank Patty Jansma, Andrea Wellington, Stephan Dong, Marija Zaruba, Mays Imad, David Tyler Eves, Jamie Ramirez, Eleazar Togawa Moreno, and Milos Babic for their technical help and critical feedback. This work was supported by grants from NINDS (R01 NS083849 to SSC (PI) and KEZ (subaward); R21 NS094809 to KEZ).

## Additional information

### Funding

| Funder | Grant reference number | Author |
| --- | --- | --- |
| National Institute of Neurological Disorders and Stroke | R01NS083849 | Sreeganga S Chandra |
| National Institute of Neurological Disorders and Stroke | R21NS094809 | Konrad E Zinsmaier |

The funders had no role in study design, data collection and interpretation, or the decision to submit the work for publication.

### Author contributions

Elliot Imler, Conceptualization, Formal analysis, Investigation, Visualization, Methodology, Writing—original draft, Writing—review and editing; Jin Sang Pyon, Selina Kindelay, Meaghan Torvund, Investigation; Yong-quan Zhang, Resources, Writing—review and editing; Sreeganga S Chandra, Conceptualization, Resources, Funding acquisition, Writing—review and editing; Konrad E Zinsmaier, Conceptualization, Resources, Data curation, Formal analysis, Supervision, Funding acquisition, Visualization, Methodology, Writing—original draft, Project administration, Writing—review and editing

### Author ORCIDs

Sreeganga S Chandra https://orcid.org/0000-0001-9035-1733
Konrad E Zinsmaier https://orcid.org/0000-0002-9992-6238

### Decision letter and Author response

Decision letter https://doi.org/10.7554/eLife.46607.018
Author response https://doi.org/10.7554/eLife.46607.019

## Additional files

### Supplementary files

• Transparent reporting form DOI: https://doi.org/10.7554/eLife.46607.016

### Data availability

All data generated or analysed during this study are included in the manuscript and supporting files.

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
