## [Decision Letter]

**Acceptance summary:**

The neuronal ceroid lipofuscinoses, the most well-known of which is Batten disease, are a genetically heterogeneous group of neurological disorders that often involve defects in lysosomal storage. In this paper, the authors focus on *CLN4*, a less-well understood, autosomal dominant, form of the disease caused by mutations in the gene for the synaptic vesicle protein CSPα. To understand how the mutant gene may cause defects in CNS development, the authors expressed CSPα mutants associated with *CLN4* in *Drosophila* neurons and used a variety of methods to investigate the consequences. The authors demonstrate that, rather than associate with synaptic vesicles, the mutant proteins form abnormal dysfunctional aggregate-like inclusions. Thus, the disease phenotypes appear to be caused by a dominant gain-of-toxicity effect associated with mislocalization and, perhaps, aggregation. Scientists interested in neurological disorders, as well as those desiring to learn how to study human diseases using model organisms, would greatly benefit from reading this paper.

**Decision letter after peer review:**

Thank you for submitting your article "A *Drosophila* model of neuronal ceroid lipofuscinosis *CLN4* reveals a hypermorphic gain of function mechanism" for consideration by *eLife*. Your article has been reviewed by two peer reviewers, and the evaluation has been overseen by a Reviewing Editor and Michael Eisen as the Senior Editor. The reviewers have opted to remain anonymous.

The reviewers have discussed the reviews with one another and the Reviewing Editor has drafted this decision to help you prepare a revised submission.

Summary:

The work presented uses a *Drosophila* model to study neuronal ceroid lipofuscinosis *CLN4*, caused by mutations in CSPα, a synaptic vesicle-associated protein. To accomplish this, they used a classical approach of expressing human *CLn4* mutants in developing flies then carefully analyzing effects on neurons and axons. They provide convincing evidence that mutant proteins oligomerize and essentially as dominant "gain of toxic function", ultimately impairing neuronal function.

The two reviewers in this paper both have expertise in disease modeling. However, they use very different systems: one is focused on the use of neurons produced from human iPSCs whereas the other is primarily a *Drosophila* biologist. Both offer similar positive remarks about the value and quality of this paper and its usefulness in providing a greater understanding of a rare and complex neurodegenerative disease. Both also offer suggestions for improving the current manuscript that are also relatively similar. Once their suggested experiments are performed, this paper should be publishable in *eLife*.

Essential revisions:

1) Overall, the authors report a set of major downstream phenotypes in their mutant *CLN4* models, including ubi-hCSPα oligomers, impaired protein homeostasis, reduced synaptic hCSPα levels and accumulation on prelysosomal endosomes. However, not all phenotypes are consistent with the gain of function (GoF) mechanism emphasized in the title and Abstract – in particular, impaired localization of mutant hCSPα (reduced synaptic levels presumably due to oligomerization/ubiquitination/re-routing) and overall increased levels of ubiquitinated proteins (possibly due to reduced CSP/Hsc70 activity). If it's unknown how these phenotypes vs. excessive oligomerization and membrane trafficking contribute to disease, this topic deserves greater clarity. The way the data is currently presented, it's difficult to connect the dots on all of the different phenotypes associated with mutant *CLN4*, and it would be helpful for the authors to add additional discussion /framing around what these different phenotypes may mean, how they may relate to each other and how they may relate to disease.

2) Some points in the Discussion are overstatements and should be adjusted. In particular, in the subsection “Dose-dependency of phenotypes in the *Drosophila CLN4* model”, the authors claim they 'model dominant inheritance'. They do not, they have controlled gene dosage as they correctly state in the next line. They compare their results to an unusual *CLN4* case study, but I do not see how this comparison supports or refutes a 'dose dependent build-up' of pathology. They have not carried out assays in *Drosophila* that establish toxicity can occur without neurodegeneration or the brain functional disruption. I would remove this statement.

3) Does reducing endogenous dCSP levels have any impact on reduced synaptic hCSPα levels or overall increased levels of ubiquitinated proteins? Similarly, if co-chaperone activity of Hsc70 contributes to accumulation on prelysosomal endosomes (Figure 8I), does loss of Hsc70 also impact protein homeostasis or reduced synaptic hCSPα levels?

4) It would be a very powerful addition to support oligomerization as the toxic mechanism if the authors could demonstrate that expression of *C(4-7)A* mutants of the cysteine string domain of hCSP in the context of *CLN4* mutations (see Diez-Ardanuy et al., 2017) are not toxic.

5) Throughout the manuscript, the authors manipulate hCSP and dCSP in various experiments (e.g. to test models of toxicity). It is therefore key that the functional comparison of hCSP with dCSP be carefully described. In particular, in Figure 1D the authors should add UAS WT dCSP2 as a control. The authors use elav-Gal4 a neural driver to express hCSP transgenes. It is difficult to evaluate how effectively WT hCSP rescues dcsp mutants using this driver without this comparison. Similarly, in Figure 4—figure supplement 2 the authors should add lifespan assays for elav-Gal4 driven expression of V117R and I118∆ dCSP2 as a comparison to hCSP mutant overexpression to strengthen their conclusion the mutant dCSP2 acts similarly to mutant hCSP. The authors already have all the necessary stocks for these experiments.

6) A useful screening assay, particularly for 'toxic proteins' in *Drosophila* is overexpression in the eye as the authors have done here. However the authors should provide quantitative scoring of the phenotypes they observe in the eye (e.g. Santa-Maria et al., 2015) to allow comparison of the effects they see in both Figure 3 and Figure 8.

7) The authors say lipidated levels of WT hCSP "were about 0.8 -1.5 times endogenous dCSP". Given the accuracy of their other data in Figure 2 can they explain why this range is so large? If necessary this experiment should be repeated to improve the comparison of lipidation of hCSP to dCSP.

---

## [Author Response]

Essential revisions:

*1) Overall, the authors report a set of major downstream phenotypes in their mutant CLN4 models, including ubi-hCSP*α *oligomers, impaired protein homeostasis, reduced synaptic hCSP*α *levels and accumulation on prelysosomal endosomes. However, not all phenotypes are consistent with the gain of function (GoF) mechanism emphasized in the title and Abstract – in particular, impaired localization of mutant hCSP*α *(reduced synaptic levels presumably due to oligomerization/ubiquitination/re-routing) and overall increased levels of ubiquitinated proteins (possibly due to reduced CSP/Hsc70 activity). If it's unknown how these phenotypes vs. excessive oligomerization and membrane trafficking contribute to disease, this topic deserves greater clarity. The way the data is currently presented, it's difficult to connect the dots on all of the different phenotypes associated with mutant CLN4, and it would be helpful for the authors to add additional discussion /framing around what these different phenotypes may mean, how they may relate to each other and how they may relate to disease.*

In response to the reviewer’s suggestion, we have revised parts of the Discussion section to better describe the likely relation of the observed phenotypes with each other.

In addition, we have revised the Discussion to clearly describe the phenotypes that support the idea that *CLN4* mutations primarily cause hypermorphic gain of function phenotypes. This section also describes phenotypes that do not fit into this classification (subsection “*CLN4* alleles genetically resemble hypermorphic gain of function mutations”). This includes 2 suggestions for the potentially primary effects of the hypermorphic *CLN4* mutations. Specifically, we suggest a potential effect either on the dimerization properties of hCSPα leading to excessive oligomerization and a potential effect on the trafficking of hCSPα onto late endosomes for MAPS.

2) Some points in the Discussion are overstatements and should be adjusted. In particular, in the subsection “Dose-dependency of phenotypes in the *Drosophila* CLN4 model”, the authors claim they 'model dominant inheritance'. They do not, they have controlled gene dosage as they correctly state in the next line.

We agree and have rephrased the sentence as follows: “To assess the dose-dependency of *CLN4* phenotypes in the fly model, we controlled the gene dosage of transgenically expressed *CLN4* mutant hCSPα by neuronally expressing either one or two mutant transgenes in the presence of two endogenous WT dCSP gene copies”.

They compare their results to an unusual CLN4 case study, but I do not see how this comparison supports or refutes a 'dose dependent build-up' of pathology. They have not carried out assays in *Drosophila* that establish toxicity can occur without neurodegeneration or the brain functional disruption. I would remove this statement.

We agree and have removed any reference to this case study.

*3) Does reducing endogenous dCSP levels have any impact on reduced synaptic hCSP*α *levels or overall increased levels of ubiquitinated proteins? Similarly, if co-chaperone activity of Hsc70 contributes to accumulation on prelysosomal endosomes (Figure 8I), does loss of Hsc70 also impact protein homeostasis or reduced synaptic hCSP*α *levels?*

We have performed the necessary experiments to answer these questions and found that reducing dCSP levels significantly decreased the levels of ubiquitinated proteins in larval brains (Figure 8C). Unexpectedly, reducing dCSP levels enhanced the already reduced synaptic protein levels of hCSP-L115 and -L116 at larval NMJs (Figure 8D).

4) It would be a very powerful addition to support oligomerization as the toxic mechanism if the authors could demonstrate that expression of C(4-7)A mutants of the cysteine string domain of hCSP in the context of CLN4 mutations (see Diez-Ardanuy et al., 2017) are not toxic.

While we agree that further evidence supporting oligomerization as the toxic mechanism would be a powerful addition to our study, we did not pursue the suggested experiment for 2 major reasons. First, even an optimistic estimate for generating and analyzing these *CLN4/C(4-7)A* double mutants will require at least 6 months, which is well beyond the suggested time frame for revisions.

In addition, after carefully considering the impact of the requested experiment, it may not provide us with strong evidence supporting oligomerization as the toxic mechanism because CSPα containing only the *C(4-7)A* mutation is only weakly associated with membranes (Greaves and Chamberlain, 2006). Consistently, *C(4-7)A* alone decreased the overall efficiency of palmitoylation of the remaining cysteines in the CS domain of CSPα rendering large fractions of the protein cytosolic (Diez-Ardanuy et al., 2017; Greaves and Chamberlain, 2006). *C(4-7)A* in *CLN4* mutant hCSPα also showed a decreased palmitoylation efficiency similar to WT CSPα (Diez-Ardanuy, 2017 #5899). Hence, the loss of aggregation of *CLN4/C(4-7)A* double mutants could have been due to a reduced membrane association rather than a specific role of the C(4-7) region in the aggregation process. To address this, Diez-Ardanuy et al. examined *CLN4/C(4-7)L* double mutants because the L substitutions preserve the membrane association of CSPα (but not palmitoylation, arresting trafficking in the ER). Since protein aggregation of *CLN4* mutant CSPα was suppressed by the *C(4-7)L* mutation, Diez-Ardanuy et al. concluded that the C(4-7) region of the CS domain is central to the aggregation process.

Accordingly, the use of *CLN4* double mutants containing *C(4-7)A* and/or *C(4-7)L* to link the oligomerization (aggregation) of *CLN4* mutant CSPα to toxicity in the fly model is problematic for a number of reasons: (1) Since *CLN4/C(4-7)A* is not efficiently palmitoylated and a large fraction is cytosolic in PC12 cells, expression of the double mutant protein in fly neurons has likely similar effects. Hence, a potential attenuation of toxicity could be due to both the expected loss of oligomerization or a reduced membrane association of the protein with SVPs or SVs. (2) Using *CLN4/C(4-7)L* double mutant in addition is unlikely to solve this problem because its trafficking is arrested in the ER (Diez-Ardanuy et al., 2017; Greaves and Chamberlain, 2006). Hence, a potential attenuation of toxicity could be due to both the expected loss of oligomerization or a reduced association of the protein with SVPs or SVs. Alternatively, the expected differential mislocalization of CSPα with only *C(4-7)A* and/or *C(4-7)L* may cause a significant degree of toxicity by itself that has a different origin than that of *CLN4* mutations. If so, this will severely limit deriving conclusions from the double mutants in regard to toxicity. This is likely to be case because mutations in fly dCSP covering the C(4-7) region affecting exhibit severe dominant effects compromising viability (Arnold, 2004 #3616).

5) Throughout the manuscript, the authors manipulate hCSP and dCSP in various experiments (e.g. to test models of toxicity). It is therefore key that the functional comparison of hCSP with dCSP be carefully described. In particular, in Figure 1D the authors should add UAS WT dCSP2 as a control. The authors use elav-Gal4 a neural driver to express hCSP transgenes. It is difficult to evaluate how effectively WT hCSP rescues dcsp mutants using this driver without this comparison.

We agree and have performed the requested rescue of *csp* deletion mutants with a WT dCSP2 transgene, and have added the data to Figure 1D.

Similarly, in Figure 4—figure supplement 2 the authors should add lifespan assays for elav-Gal4 driven expression of V117R and I118∆ dCSP2 as a comparison to hCSP mutant overexpression to strengthen their conclusion the mutant dCSP2 acts similarly to mutant hCSP. The authors already have all the necessary stocks for these experiments.

We agree and have performed the requested viability assays. Single copy expression of dCSP2, dCSP-V117 or -I118Δ had no effect on viability during development at 23ºC (not shown) and 27ºC (Figure 4—figure supplement 1A). Adult lifespan at 27ºC was also normal (Figure 4—figure supplement 1D). Two copy expression reduced the viability of animals expressing WT dCSP or dCSP-V117 to a similar degree (Figure 4—figure supplement 1B) The few surviving adult WT dCSP and dCSP-V117 animals exhibited rough eyes, abnormally inflated wings, impaired locomotion and died within a day. Adult flies expressing dCSP-I118 were never observed (Figure 4—figure supplement 1B).

6) A useful screening assay, particularly for 'toxic proteins' in *Drosophila* is overexpression in the eye as the authors have done here. However the authors should provide quantitative scoring of the phenotypes they observe in the eye (e.g. Santa-Maria et al., 2015) to allow comparison of the effects they see in both Figure 3 and Figure 8.

We agree and have performed the requested semi-quantitative assessment of the eye phenotype. The data are shown in Figures 3C and 9E and discussed in the respective Results section. The assessment is explained in the Materials and methods section (subsection “Quantification of Eye Phenotypes”).

7) The authors say lipidated levels of WT hCSP "were about 0.8 -1.5 times endogenous dCSP". Given the accuracy of their other data in Figure 2 can they explain why this range is so large? If necessary this experiment should be repeated to improve the comparison of lipidation of hCSP to dCSP.

The data were obtained by measuring staining intensities on Western Blots to a dilution ladder of purified recombinant hCSPα and dCSP. These values were then compared after normalizing them to the input level. The stated range was due to different values obtained when comparing to two closest purified protein bands for each species. The observed range may be due to experimental variation in which case averaging would be appropriate. However, it could be also due to non-linear antibody/antigen interactions due to the large range of protein inputs used for the ladder. To address this issue, we have revised the experiment and now provide the average value (0.98 ± 0.23; subsection “*CLN4* mutations cause formation of SDS-resistant and ubiquitinated hCSPα protein oligomers in neurons”, second paragraph) instead of the maximal range.